# DiFA: Inference-Time Forward-Process Alignment for Diffusion Models

**Shigui Li** [1]   **Delu Zeng** [2]

## Abstract

The prevailing inference framework for diffusion models formulates generation fundamentally as a problem of numerical integration. This perspective casts the model as an exact estimator, neglecting the inherent statistical uncertainty of the denoising process. In this work, we propose Forward-Process Aligned Diffusion prediction (**DiFA**), a training-free framework that reframes inference-time data prediction refinement as a sequential state estimation problem. Rather than reusing past outputs solely for numerical integration, DiFA treats iterative data predictions along the reverse trajectory as correlated observations to build a forward-aligned temporal consensus. Inspired by Kalman filtering, this consensus aggregates historical predictions according to structural consistency and noise-level compatibility. To counteract the over-smoothing tendency of temporal consensus, we introduce a deviation guidance mechanism to adaptively preserve residual details. Empirically, DiFA yields significant improvements on CIFAR-10 and ImageNet across the evaluated metrics, including FID, IS, and FD-DINOv2, demonstrating that aligning inference with the forward statistical structure substantially improves generative fidelity.

## 1. Introduction

Diffusion models (DMs) have demonstrated striking success in high-fidelity generative tasks (Sohl-Dickstein et al., 2015; Ho et al., 2020; Song et al., 2021c; Dhariwal & Nichol, 2021), spanning from photorealistic image synthesis to complex scientific simulations (Rombach et al., 2022; Blattmann et al., 2023; Podell et al., 2024; Batifol et al., 2025). They leverage an iterative denoising process to progressively

[1]School of Mathematics, South China University of Technology, Guangzhou, China; [2]School of Electronic and Information Engineering, South China University of Technology, Guangzhou, China. Correspondence to: Delu Zeng <dlzeng@scut.edu.cn>.

*Proceedings of the 43rd International Conference on Machine Learning*, Seoul, South Korea. PMLR 306, 2026. Copyright 2026 by the author(s).

transform unstructured noise into samples from the target data distribution (Kingma et al., 2021; Karras et al., 2022; 2024b). However, this iterative process inherently imposes a significant inference burden (Lu et al., 2022; Zhao et al., 2023; Wimbauer et al., 2024; Luo et al., 2024). Unlike one-step generative paradigms (e.g., GANs (Goodfellow et al., 2014), VAEs (Kingma, 2013)), DMs rely on a substantial number of network evaluations (NFEs) to progressively traverse the data manifold and generate high-fidelity samples (Ho et al., 2020; Farghly et al., 2025).

The challenge in this diffusion paradigm lies in reducing sampling steps without compromising sample quality (Liu et al., 2022; Bao et al., 2022; Zhang & Chen, 2023; Zheng et al., 2023; Li et al., 2023; Ma et al., 2024a; Sabour et al., 2024; Lu et al., 2025; Jiang et al., 2025; Li et al., 2025; Fu et al., 2025; Ma et al., 2025). Standard samplers, whether deterministic (ODE-based) or stochastic (SDE-based), typically rely on local linear approximations of the score function to predict the next state (Lu et al., 2022; Gonzalez et al., 2023; Ye et al., 2024; Abuduweili et al., 2025; Tang et al., 2025). We argue that this linearized perspective is inherently geometrically agnostic. In high-curvature regions, such approximations introduce inevitable estimation biases (Karras et al., 2022; Esser et al., 2024; Zhang et al., 2025). Furthermore, although injected noise is necessary for diversity, it introduces a per-step variance that is severely amplified during discrete, few-step inference (Song et al., 2021a). These errors are not isolated but cumulative (Ning et al., 2024; Chang et al., 2026; Li & Zeng, 2026), with minor deviations compounding into irreversible prediction drift.

To address these challenges, research efforts have largely consolidated into two orthogonal directions, each with distinct limitations. On one hand, distillation techniques attempt to compress the trajectory into fewer steps (Salimans & Ho, 2022; Song et al., 2023; Meng et al., 2023; Liu et al., 2023a; Frans et al., 2025). While reducing inference latency, they incur substantial retraining overhead and are often confined to task-specific customizations (Luo et al., 2023; Liu et al., 2024; Yin et al., 2024; Sauer et al., 2024; Dong et al., 2024; Shen et al., 2025; Sabour et al., 2025). This may restrict the model to a narrowed distribution and compromise the zero-shot generalization and diversity of the original teacher model. On the other hand, advanced solvers employ higher-order numerical methods to reduce

*Figure 1. DiFA Mechanism of Inference-Time Forward-Process Alignment.* Traditional samplers (Top) use each denoiser prediction only for the current solver step, discarding temporal redundancy along the trajectory. DiFA (Bottom) constructs a forward-aligned temporal consensus $\hat{\boldsymbol{x}}_0^{\mathrm{cons}}(t)$ from historical predictions and uses it as an anchor for residual construction. Specifically, $\boldsymbol{r}_t = \hat{\boldsymbol{x}}_0^{(t)} - \hat{\boldsymbol{x}}_0^{\mathrm{cons}}(t)$ is transformed into deviation guidance $\boldsymbol{g}_t = \mathrm{G}(\boldsymbol{r}_t)$, yielding the refined prediction $\hat{\boldsymbol{x}}_0^{\mathrm{DiFA}}(t) = \hat{\boldsymbol{x}}_0^{(t)} + \omega\boldsymbol{g}_t$ without additional NFEs.

truncation errors (Lu et al., 2022). However, by treating the fixed model as an exact estimator for correcting temporal discretization errors, they follow each instantaneous prediction as if it were a sufficiently accurate clean-signal estimate, leaving its uncertainty and the temporal redundancy of the prediction sequence unexploited.

Inspired by EDM2 (Karras et al., 2024a), which suggests DMs can self-correct using just *bad versions* of themselves, we identify the bottleneck not merely time discretization error, but the intrinsic estimation uncertainty of the learned model. Unlike high-order ODE solvers that assume the vector field is ground truth and use auxiliary points solely to refine temporal integration, we argue that the learned score is an imperfect, smoothed approximation of the posterior mean due to the MSE training objective. Consequently, standard samplers treat each prediction as a transient estimation, blindly following a biased tangent while discarding the rich information embedded in the sampling trajectory. *Can we then harvest the latent consensus from the sequence of historical predictions to rectify this estimation drift without any additional network evaluations?* By synthesizing information across the temporal sequence, the model can leverage its own historical redundancy to mitigate linearization biases. The base DM thus holds untapped potential; it merely lacks a mechanism to unleash this intrinsic self-guidance through temporal consensus. Guided by this insight, we propose Forward-Process Aligned Diffusion prediction (DiFA), a principled inference-time algorithm that rectifies signal predictions via forward-process alignment. DiFA does not replace the numerical solver. Instead, it refines the clean prediction supplied to each solver step. Motivated by Kalman

estimation under an idealized static-anchor model, DiFA implements this principle through forward-aligned sliding-window temporal consensus, where historical predictions are fused according to structural consistency and noise-level compatibility. To avoid over-smoothing, DiFA further constructs a deviation guidance direction and modulates its smoothed and high-frequency components.

In summary, our contributions are as follows. First, we identify clean-signal prediction drift during diffusion inference as arising from time-dependent estimation uncertainty, posterior ambiguity, and trajectory mismatch, shifting the focus from solely improving numerical integration to refining the clean-signal estimate provided to downstream solvers. Second, we propose DiFA, a solver-compatible inference-time output-alignment framework that exploits temporal redundancy through adaptive temporal consensus, where historical predictions are fused according to structural consistency and noise-level-dependent reliability. Third, motivated by Best Linear Unbiased Estimation (BLUE), we introduce a principled approximation to effectively mitigate the aforementioned prediction uncertainty. Extensive evaluations on CIFAR-10 and ImageNet show that DiFA consistently improves existing samplers across FID, IS, and FD-DINOv2 scores without additional network evaluations, demonstrating the effectiveness of correcting clean-signal predictions within the inference pipeline.

## 2. Related Work

**Accelerated Generation.** The substantial computational burden of DMs has catalyzed a broad spectrum of acceler-

ation strategies across distinct operational paradigms. At the architecture level, latent DMs (Rombach et al., 2022) improve efficiency by operating in compressed latent spaces. At the training level, researchers accelerate generation by retraining or distilling models to shorten sampling chains. Representative modeling approaches include progressive distillation (Salimans & Ho, 2022), flow matching (FM) (Lipman et al., 2023), reflow (Liu et al., 2023a), and recently developed one-step or few-step methods, such as consistency models (Song et al., 2023), adversarial distillation (Sauer et al., 2024), shortcut models (Frans et al., 2025), and mean flows (Geng et al., 2025). Alternatively, training-free methods optimize the inference process itself. Following the deterministic formulation (Song et al., 2021a), a variety of advanced solvers have emerged. Early solvers like PNDM (Liu et al., 2022) introduced the use of historical gradients, while the DPM-Solver family (Lu et al., 2022; 2025) later proposed specialized exponential integrators for diffusion ODEs. Specialized solvers with lightweight learning have been explored to bridge the gap between both paradigms (Tong et al., 2025). Moreover, complementary strategies include schedule optimization (Sabour et al., 2024), parallel decoding (Shih et al., 2023) and caching mechanisms (Wimbauer et al., 2024; Ma et al., 2024b). While effective in reducing NFEs, these methods prioritize temporal precision, often neglecting the intrinsic estimation variance.

**Inference-Time Rectification.** A pivotal challenge in diffusion model inference lies in mitigating trajectory instability and statistical discrepancies (Lin et al., 2024). Accordingly, refinement methodologies have progressed from variance-controlled stochastic sampling (Ho et al., 2020; Song et al., 2021c; Bao et al., 2022) to explicit rectification. While advanced solvers like UniPC (Zhao et al., 2023) emphasize numerical accuracy for efficient integration, recent approaches like Zigzag (Bai et al., 2025) and inference-time scaling strategies (Ma et al., 2025) explicitly amend intermediate errors or optimize search paths, often trading computational efficiency for improved generation quality. Recently, to overcome the rigidity of conventional numerical solvers, EVODiff (Li et al., 2025) rectifies the inference path via entropy-aware variance optimization. Although classical antithetic sampling (Ren et al., 2019) has been revisited recently for initial-noise pairing (Jia et al., 2025) to improve global diversity, its capacity for dynamic trajectory rectification remains largely unexplored. Existing strategies typically rely on external task-specific constraints (Chung et al., 2022; Bansal et al., 2024; Yang et al., 2024), heuristic interventions on internal structures (Ho & Salimans, 2022; Hong et al., 2023; Epstein et al., 2023; Liu et al., 2023b; Shen et al., 2024; Ahn et al., 2024), or temporal rescaling (Park et al., 2025). However, these methods mainly improve semantic consistency while leaving the inherent estimation variance and high-frequency degradation caused by MSE-based denoising unaddressed. Although EDM2 (Karras et al., 2024a) introduces self-guidance, it still relies on auxiliary degraded models and hand-crafted guidance cues. In contrast, DiFA offers an intrinsic and training-free framework grounded in classical estimation principles. It treats denoising as a filtering process, avoiding reliance on external models or heuristic guidance.

**Limitations and Our Contribution.** Existing inference strategies face a fundamental dilemma: distillation incurs costly retraining and potential quality degradation, whereas standard solvers neglect the estimation variance and detail attenuation of MSE-trained denoisers. We propose Forward-Process Aligned Diffusion prediction (*DiFA*), a training-free framework that exploits temporal redundancy to refine clean-signal predictions. Unlike solvers that use historical model evaluations to refine numerical integration, DiFA constructs a forward-aligned reference in the clean-prediction space before the unchanged solver update, without requiring auxiliary models. Under an idealized independent-view model, the BLUE analysis yields a principled fusion rule for historical predictions, which motivates the practical forward-aligned consensus. Deviation guidance is designed to counteract the detail attenuation introduced by temporal aggregation. By aligning reverse inference with the forward statistical structure, DiFA mitigates prediction drift and better exploits the capacity of pretrained generative models.

## 3. Preliminaries

### 3.1. Diffusion Models and Probability Flow

Diffusion models (DMs) rely on a constructed forward process that progressively corrupts data $\boldsymbol{x}_0 \sim q(\boldsymbol{x}_0)$ into Gaussian noise (Ho et al., 2020). The marginal distribution of a noisy sample $\boldsymbol{x}_t$ at time $t$ can generally be expressed as:

$$q(\boldsymbol{x}_t | \boldsymbol{x}_0) = \mathcal{N}(\boldsymbol{x}_t; \alpha_t \boldsymbol{x}_0, \sigma_t^2 \boldsymbol{I}), \quad (1)$$

where $\alpha_t$ and $\sigma_t$ denote the signal and noise scale parameters, respectively. The model learns to reverse this process by predicting the noise component $\boldsymbol{\epsilon}$ via a network $\boldsymbol{\epsilon}_\theta(\boldsymbol{x}_t, t)$, optimized by minimizing the mean squared error (MSE):

$$\mathrm{L} = \mathbb{E}_{t, \boldsymbol{x}_0, \boldsymbol{\epsilon}} \| \boldsymbol{\epsilon} - \boldsymbol{\epsilon}_{\boldsymbol{\theta}}(\alpha_t \boldsymbol{x}_0 + \sigma_t \boldsymbol{\epsilon}, t) \|^2. \quad (2)$$

While classic DDPMs operate on a discrete schedule, the transition kernel can be generalized to a continuous-time setting (Kingma et al., 2021). This extension unifies the framework under a stochastic differential equation (SDE) formulation (Song et al., 2021c), where the generative process follows the reverse-time dynamics:

$$\mathrm{d}\,\boldsymbol{x} = [f(t)\boldsymbol{x} - g(t)^2 \nabla_{\boldsymbol{x}} \log p_t(\boldsymbol{x})]\,\mathrm{d}\,t + g(t)\,\mathrm{d}\,\bar{\boldsymbol{w}}, \quad (3)$$

where $\mathrm{d}\,\bar{\boldsymbol{w}}$ is the standard Wiener process in reverse time. Crucially, there exists a deterministic Probability Flow ODE

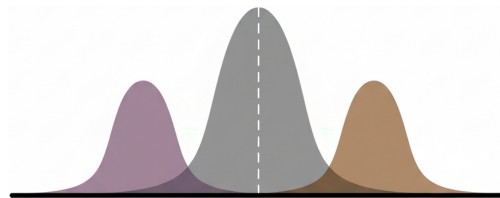

*Figure 2.* Illustrating the mean-field discrepancy in model estimation through spatial shift and magnitude scaling.

(PF-ODE) that shares the same marginal as the SDE:

$$\frac{\mathrm{d}\,\boldsymbol{x}}{\mathrm{d}\,t} = f(t)\boldsymbol{x} - \frac{1}{2}g(t)^2 \nabla_{\boldsymbol{x}} \log p_t(\boldsymbol{x}). \quad (4)$$

### 3.2. Inference Framework and Linearization Bias

As the score function $\nabla_{\boldsymbol{x}} \log p_t(\boldsymbol{x})$ is parameterized as $-\boldsymbol{\epsilon}_\theta(\boldsymbol{x}_t, t)/\sigma_t$ (Song et al., 2021c; Karras et al., 2022) and the prediction relationship of $\boldsymbol{x}_{\boldsymbol{\theta}}(\boldsymbol{x}_t, t) = \frac{\boldsymbol{x}_t - \sigma_t \boldsymbol{\epsilon}_{\boldsymbol{\theta}}(\boldsymbol{x}_t, t)}{\alpha_t}$ (Kingma et al., 2021), the inference ODE is formulated as

$$\frac{\mathrm{d}\,\boldsymbol{x}}{\mathrm{d}\,t} = \left( f(t) + \frac{g^2(t)}{2\sigma_t^2} \right) \boldsymbol{x}_t - \alpha_t \frac{g^2(t)}{2\sigma_t^2} \boldsymbol{x}_{\boldsymbol{\theta}}(\boldsymbol{x}_t, t). \quad (5)$$

This framework is generalized by the EDM formulation (Karras et al., 2022; 2024a), which stabilizes probability flow via signal scaling. A pivotal innovation in EDM is the introduction of input preconditioning, which explicitly standardizes the network input into a *signal-centric form*:

$$\frac{\boldsymbol{x}_t}{s(t)} = \boldsymbol{x}_0 + \sigma(t)\boldsymbol{\epsilon}. \quad (6)$$

By normalizing the signal scale to unity across all time $t$, this formulation reveals that the diffusion trajectory effectively orbits a static anchor $\boldsymbol{x}_0$. This architectural alignment of the input state to $\boldsymbol{x}_0$ resonates with the core intuition of our work, where we seek to align the output predictions to the same statistical invariant. This concept of straightening the generative path is also central to concurrent works on FMs (Lipman et al., 2023; Liu et al., 2023a). In fact, FM can be seen as a special case of diffusion modeling (Kingma & Gao, 2023), which typically defines the process as a linear interpolation $\boldsymbol{x}_t = (1-t)\boldsymbol{x}_0 + t\boldsymbol{\epsilon}$. While trajectory straightness is widely regarded as a key property for minimizing discretization errors, the perspective of rectified diffusion (Wang et al., 2025) suggests that pre-defined linearity is insufficient. Instead, the efficacy of rectification stems from bootstrap retraining, which aligns the model with ODE coupling of a pre-trained teacher. This process facilitates the transport curvature, allowing first-order solvers to achieve one-step generation (Frans et al., 2025; Geng et al., 2025).

In this work, we focus on the general EDM formulation without such expensive retraining. Consequently, our standard pre-trained models do not strictly satisfy the first-order

property. To quantify the impact of this, let $\Phi(\boldsymbol{x}, t)$ denote the drift vector field defined by the RHS of Eq. (5). Standard inference numerically integrates this ODE by approximating the trajectory as locally linear within a step $\Delta t$ (e.g., a first-order Euler step $\boldsymbol{x}_{t-1} \approx \boldsymbol{x}_t + \Delta t \cdot \Phi(\boldsymbol{x}_t, t)$). However, due to the lack of retraining, the generative manifold retains significant curvature. This geometric complexity, coupled with the inherent estimation variance of the model $\Phi$, renders the trajectory highly sensitive to local prediction errors. Standard solvers, which blindly follow the instantaneous tangent, accumulate these errors as geometric drift.

## 4. Method

We propose Forward-Process Aligned Diffusion prediction (DiFA), a theory-inspired and principled algorithm for inference-time clean-signal prediction alignment. DiFA is a training-free framework and does not replace existing numerical solvers. Instead, it operates at the interface between the pretrained denoiser and the downstream solver by refining the data prediction supplied to each solver step. From the perspective of the forward process of diffusion models, the central idea is to reframe the inference-time predictions along the reverse trajectory as temporally related observations of a locally stable clean-signal anchor. DiFA uses the common-anchor geometry and the noise-level-dependent reliability structure induced by the forward process to filter these multi-step predictions before they are passed to the downstream solver.

### 4.1. Diffusion Sampling as Sequential Estimation

During inference, the denoiser produces a sequence of clean-signal predictions,

$$\hat{\boldsymbol{x}}_0^{(t_i)} = D_\theta(\boldsymbol{x}_{t_i}, t_i), \qquad i = 1, \ldots, N. \quad (7)$$

Assuming that reverse inference proceeds in the order $t_N > t_{N-1} > \cdots > t_0$, for a current reverse step $t_i$, let

$$\mathrm{W}_K(t_i) \subseteq \{t_j : j > i\}, \qquad |\mathrm{W}_K(t_i)| \leq K, \quad (8)$$

denote a causal temporal window of recently processed historical steps, and define

$$\mathrm{W}_K^+(t_i) = \mathrm{W}_K(t_i) \cup \{t_i\}. \quad (9)$$

Rather than treating the predictions in this local neighborhood as isolated one-step estimates, we view them as temporally correlated observations of a locally stable trajectory-implied clean anchor. For $t_j \in \mathrm{W}_K^+(t_i)$, we write

$$\hat{\boldsymbol{x}}_0^{(t_j)} = \boldsymbol{x}_{0,\star}^{(t_i)} + \boldsymbol{b}_{j|i} + \boldsymbol{\xi}_{j|i}, \; t_j \in \mathrm{W}_K^+(t_i), \quad (10)$$

where $\boldsymbol{x}_{0,\star}^{(t_i)}$ denotes the trajectory-implied clean anchor, assumed locally stable within this neighborhood, while $\boldsymbol{b}_{j|i}$

and $\boldsymbol{\xi}_{j|i}$ capture the systematic bias and residual uncertainty with respect to this anchor, respectively.

Due to distribution shifts in few-step trajectories, predictions from MSE-trained denoisers may exhibit systematic bias and temporal correlation. The reliability ordering used by DiFA is instead induced by the normalized forward process. From Eq. (1), the forward process admits the equivalent signal-normalized representation,

$$\bar{\boldsymbol{x}}_t := \frac{\boldsymbol{x}_t}{\alpha_t} = \boldsymbol{x}_0 + \frac{\sigma_t}{\alpha_t}\boldsymbol{\epsilon} = \boldsymbol{x}_0 + \mathrm{SNR}(t)^{-1/2}\boldsymbol{\epsilon}, \quad (11)$$

where $\boldsymbol{\epsilon} \sim \mathcal{N}(\boldsymbol{0}, \boldsymbol{I})$. Accordingly,

$$\bar{\boldsymbol{x}}_t \mid \boldsymbol{x}_0 \sim \mathcal{N}\left(\boldsymbol{x}_0, \mathrm{SNR}(t)^{-1}\boldsymbol{I}\right). \quad (12)$$

Therefore, in the forward-corruption space, states at different noise levels share the same clean-data anchor ($\boldsymbol{x}_0$), and $\mathrm{SNR}(t)$ exactly characterizes the conditional observation precision of the corresponding forward observation.

This forward statistical geometry motivates DiFA's treatment of the reverse-trajectory predictions: although learned denoiser errors may be biased and temporally correlated rather than exactly following the forward observation covariance, the prediction sequence can still be organized and filtered according to the common-anchor structure and the noise-level reliability ordering induced by the forward process. While existing numerical solvers widely utilize historical outputs to refine temporal integration, they fail to explicitly and dynamically construct a forward-aligned reference anchor to rectify prediction drift during inference.

### 4.2. Forward-Process Alignment Principle

Based on the forward-induced common-anchor geometry established above, we now formalize the alignment principle underlying DiFA in Algorithm 1.

**Definition 4.1** (Forward-Process Alignment). Let $\{\hat{\boldsymbol{x}}_0^{(t_i)}\}_{i=1}^N$ denote the data predictions along a reverse trajectory. Forward-Process Alignment is the principle that inference-time prediction refinement should be constructed relative to the common-anchor geometry and noise-level reliability ordering induced by the forward noising process.

This principle is instantiated by first constructing a temporally stable reference anchor in the clean-prediction space and then using the anchor-relative deviation for current-step refinement. Accordingly, forward-process alignment specifies the reference coordinate and reliability organization for prediction refinement. To derive a canonical consensus rule implied by this principle, we introduce an idealized and tractable forward-aligned independent-view observation model. This model preserves the common-anchor geometry and inverse-SNR reliability structure of the normalized forward process, while serving as an analytical proxy rather than an exact model of the learned denoiser errors:

$$\boldsymbol{y}_i = \boldsymbol{x}_{0,\star} + \boldsymbol{\eta}_i,$$
$$\boldsymbol{\eta}_i \overset{\mathrm{ind}}{\sim} \mathcal{N}(\boldsymbol{0}, R_i \boldsymbol{I}), \quad R_i = \frac{c}{\mathrm{SNR}(t_i)}, \quad (13)$$

where $c > 0$ is a common scale constant and $\boldsymbol{x}_{0,\star}$ denotes the static anchor in this ideal analytical model. It is the idealized counterpart of the locally stable trajectory-implied anchor $\boldsymbol{x}_{0,\star}^{(t_i)}$ in Eq. (10). This model provides a canonical analytical basis for deriving the consensus estimator.

Then, we seek a linear unbiased estimator of $\boldsymbol{x}_{0,\star}$ whose weights minimize the isotropic estimation covariance:

$$\hat{\boldsymbol{x}}_{0,\mathrm{ideal}}^\star = \sum_{i=1}^n w_i \boldsymbol{y}_i, \quad \sum_{i=1}^n w_i = 1,$$
$$\{w_i^\star\}_{i=1}^n = \underset{\sum_{i=1}^n w_i = 1}{\arg\min} \sum_{i=1}^n w_i^2 R_i. \quad (14)$$

This yields the precision-weighted estimator

$$\hat{\boldsymbol{x}}_{0,\mathrm{ideal}}^\star = \frac{\sum_{i=1}^n \mathrm{SNR}(t_i)\boldsymbol{y}_i}{\sum_{i=1}^n \mathrm{SNR}(t_i)}, \quad (15)$$

where the SNR-dependent weights follow from the inverse-SNR covariance form of the forward-aligned observation.

**Proposition 4.2** (Variance Reduction under Ideal Forward-Aligned Observations). *Under the observation model in Eq. (13), the fused anchor estimator satisfies*

$$\mathrm{Cov}\left(\hat{\boldsymbol{x}}_{0,\mathrm{ideal}}^\star\right) = \left(\sum_{i=1}^n R_i^{-1}\right)^{-1}\boldsymbol{I} = \frac{c}{\sum_{i=1}^n \mathrm{SNR}(t_i)}\boldsymbol{I}, \quad (16)$$

*which is strictly smaller, in the positive-semidefinite ordering, than the covariance of any individual observation for $n \geq 2$. The proof is provided in Appendix A.1.*

This canonical anchor estimator can be reformulated as the recursive Kalman realization described in the next section.

### 4.3. Connection to Kalman Filtering

The anchor estimator in Eq. (15) admits an equivalent recursive realization through static-state Kalman updates. Under the independent-view, each observation provides precision-weighted evidence about the shared anchor, allowing the full-history estimator to be implemented recursively.

**Theorem 4.3** (Recursive Kalman Equivalence under the Ideal Independent-View Model). *Under the ideal independent-view observation model in Eq. (13), initialize the static-state recursive estimator by*

$$\hat{\boldsymbol{x}}_{0,1}^{\mathrm{rec}} = \boldsymbol{y}_1, \qquad p_1 = \frac{c}{\mathrm{SNR}(t_1)}.$$

Baseline Solver      DiFA Refined Prediction

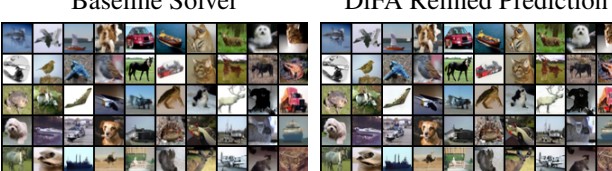

*Figure 3.* Visual quality comparison on CIFAR-10 using pre-trained EDM weights. Compared to the baseline solver, incorporating our DiFA framework significantly enhances visual fidelity and edge sharpness under identical few-step inference budgets.

For $i = 2, \ldots, n$, *assimilate* $\boldsymbol{y}_i$ *using the static-state Kalman update with observation covariance* $c\,\mathrm{SNR}(t_i)^{-1}\boldsymbol{I}$. *Then*

$$\hat{\boldsymbol{x}}_{0,n}^{\mathrm{rec}} = \frac{\sum_{i=1}^{n} \mathrm{SNR}(t_i)\boldsymbol{y}_i}{\sum_{i=1}^{n} \mathrm{SNR}(t_i)} = \hat{\boldsymbol{x}}_{0,\mathrm{ideal}}^{\star}. \quad (17)$$

*The proof is provided in Appendix A.2.*

Theorem 4.3 shows that the ideal forward-aligned anchor estimator admits an exact recursive realization through static-state Kalman updates under the independent-view model. Motivated by this result, DiFA constructs a causal historical consensus in practical reverse inference using structural and noise-level compatibility to account for deviations from the ideal model. The current prediction is then refined relative to this consensus anchor.

### 4.4. Forward-Aligned Causal Temporal Consensus

Guided by the forward-aligned anchor interpretation established above, DiFA constructs a practical reference anchor from recently processed clean-signal predictions. We parameterize the noise level using logSNR, $\ell_i = \log \mathrm{SNR}(t_i) = \log \frac{\alpha_{t_i}^2}{\sigma_{t_i}^2}$. As a monotonic reparameterization of SNR, logSNR preserves the forward-induced reliability ordering while providing improved numerical stability when comparing predictions across a wide dynamic range.

At step $t_i$, DiFA maintains the causal history buffer:

$$\mathrm{H}_K(t_i) = \left\{ \left( \hat{\boldsymbol{x}}_0^{(t_j)}, \ell_j \right) \right\}_{t_j \in \mathrm{W}_K(t_i)}. \quad (18)$$

Only previously processed predictions are aggregated as historical reference values, so that the current prediction remains available for subsequent anchor-relative refinement. The temporal consensus is defined as:

$$\hat{\boldsymbol{x}}_0^{\mathrm{cons}}(t_i) = \sum_{t_j \in \mathrm{W}_K(t_i)} a_{ij}\tilde{\boldsymbol{x}}_0^{(t_j)}, \quad \sum_{t_j \in \mathrm{W}_K(t_i)} a_{ij} = 1, \quad (19)$$

where $a_{ij} \geq 0$, $\tilde{\boldsymbol{x}}_0^{(t_j)}$ denotes an optionally aligned historical prediction. The normalized weights $a_{ij}$ favor historical candidates that are structurally compatible with the current prediction and compatible with its noise level in the

**Algorithm 1** DiFA: Forward-Process Aligned Inference for Diffusion Models

---

**Require:** $D_\theta$, downstream solver, $\{t_N, \ldots, t_0\}$
**Require:** Window size $K$, guidance strength $\omega$
1: Initialize history buffer $\mathrm{H}_K \leftarrow \emptyset$
2: Sample initial noise $\boldsymbol{x}_{t_N} \sim \mathcal{N}(\mathbf{0}, \boldsymbol{I})$
3: **for** $i = N, \ldots, 1$ **do**
4:      $\hat{\boldsymbol{x}}_0^{(t_i)} \leftarrow D_\theta(\boldsymbol{x}_{t_i}, t_i)$
5:      $\ell_i \leftarrow \log \mathrm{SNR}(t_i)$
6:      **if** $\mathrm{H}_K$ is empty **then**
7:         $\hat{\boldsymbol{x}}_0^{\mathrm{DiFA}}(t_i) \leftarrow \hat{\boldsymbol{x}}_0^{(t_i)}$
8:      **else**
9:         Construct reference anchor $\hat{\boldsymbol{x}}_0^{\mathrm{cons}}(t_i)$ using Eq. (19)
10:         $\boldsymbol{r}_{t_i} \leftarrow \hat{\boldsymbol{x}}_0^{(t_i)} - \hat{\boldsymbol{x}}_0^{\mathrm{cons}}(t_i)$
11:         Construct deviation guidance $\boldsymbol{g}_{t_i}$ using Eq. (21)
12:         $\hat{\boldsymbol{x}}_0^{\mathrm{DiFA}}(t_i) \leftarrow \hat{\boldsymbol{x}}_0^{(t_i)} + \omega\boldsymbol{g}_{t_i}$
13:      **end if**
14:      $\boldsymbol{x}_{t_{i-1}} \leftarrow \mathrm{SolverStep}\left( \boldsymbol{x}_{t_i}, \hat{\boldsymbol{x}}_0^{\mathrm{DiFA}}(t_i), t_i, t_{i-1} \right)$
15:      Update $\mathrm{H}_K$ with $(\hat{\boldsymbol{x}}_0^{(t_i)}, \ell_i)$
16: **end for**
     Return $\boldsymbol{x}_{t_0}$

---

logSNR coordinate. The current prediction is used as a query for reference construction, but is not itself aggregated into the reference anchor. This exclusion prevents trivial self-reinforcement and preserves a non-degenerate anchor-relative deviation for subsequent refinement.

The resulting consensus serves as a causal historical reference anchor in the clean-prediction space. The specific alignment and compatibility-weighting rules used in our experiments are provided in Appendix A.3.

### 4.5. Anchor-Relative Deviation Guidance

The temporal consensus acts as a multi-step filtered reference that captures information stable across recent reverse steps. Directly substituting this filtered reference for the instantaneous prediction, however, may attenuate components expressed more strongly at the current step. DiFA therefore uses the consensus as a reference anchor and defines the anchor-relative deviation:

$$\boldsymbol{r}_{t_i} = \hat{\boldsymbol{x}}_0^{(t_i)} - \hat{\boldsymbol{x}}_0^{\mathrm{cons}}(t_i). \quad (20)$$

Because this deviation may contain both informative components and unstable fluctuations, DiFA applies a noise-level-conditioned gain modulation operator:

$$\boldsymbol{g}_{t_i} = \mathrm{G}\left( \boldsymbol{r}_{t_i}, \hat{\boldsymbol{x}}_0^{(t_i)}, \ell_i \right), \quad (21)$$

where $\mathrm{G}(\cdot)$ is designed to suppress unreliable deviation components while retaining current-step refinement.

The refined clean-signal prediction is given by:

$$\hat{\boldsymbol{x}}_0^{\text{DiFA}}(t_i) = \hat{\boldsymbol{x}}_0^{(t_i)} + \omega \boldsymbol{g}_{t_i}, \ \omega \geq 0. \quad (22)$$

Thus, DiFA performs anchor-relative additive refinement: the temporal consensus defines the reference coordinate, while the modulated deviation provides the correction direction. In the lightweight instantiation evaluated in this work, $G(\cdot)$ preserves informative residual-oriented components by suppressing prediction-parallel magnitude variation and modulating spatial-frequency deviation components according to the current logSNR. Its complete formulation and all hyperparameter settings are detailed in Appendix A.3.

### 4.6. Algorithm and Discussion

Algorithm 1 summarizes the complete DiFA procedure. At each timestep, the pretrained denoiser first produces an instantaneous clean-signal prediction $\hat{\boldsymbol{x}}_0^{(t_i)}$. DiFA then refines this prediction through forward-aligned causal temporal consensus and anchor-relative deviation guidance. The corrected prediction $\hat{\boldsymbol{x}}_0^{\text{DiFA}}(t_i)$ is finally passed to the downstream solver. Therefore, DiFA does not modify the numerical integration rule itself. Instead, it improves the clean prediction consumed by the solver.

DiFA is solver-compatible because it only changes the clean-signal estimate supplied to the solver. Any solver whose update can be expressed using a clean prediction, or converted to an equivalent clean-prediction parameterization, can use DiFA without changing its integration formula. The method introduces no additional network evaluations. Its extra cost comes only from maintaining a small prediction buffer, computing local similarity weights, and applying lightweight residual operations. With a fixed window size $K$, the additional cost is $O(Kd)$ per step for a prediction of dimension $d$, which is negligible compared with NFE.

## 5. Experiments

### 5.1. Settings

**Datasets & Metrics.** We evaluate DiFA on widely adopted benchmarks: CIFAR-10 ($32 \times 32$) and ImageNet-64 ($64 \times 64$). To ensure a holistic assessment, we employ three complementary metrics: Fréchet Inception Distance (FID) (Heusel et al., 2017) for distributional fidelity, Inception Score (IS) (Salimans et al., 2016) for sample diversity and clarity, and FD-DINOv2 (Stein et al., 2023) for perceptually aligned quality assessment.

**Implementation Details.** We use official pretrained diffusion checkpoints and evaluate DiFA with DDIM (Song et al., 2021a), DPM-Solver++ (Lu et al., 2025), UniPC (Zhao et al., 2023), and Heun under the corresponding VP/logSNR and EDM parameterizations (Karras et al., 2022). All experiments are conducted on NVIDIA RTX 4090 GPUs, except

*Table 1.* Quantitative results on CIFAR-10 ($32 \times 32$). We compare DiFA against state-of-the-art GANs and diffusion models. DiFA (Ours) achieves competitive performance comparable to distillation methods (e.g., CTM) while requiring zero training. †: Methods requiring training/distillation.

| Type | Model | #Params | NFE | FID ↓ |
|---|---|---|---|---|
| GAN† | StyleGAN2-ADA (Karras et al., 2020) | 20M | 1 | 2.92 |
| | StyleGAN-XL (Sauer et al., 2022) | 18M | 1 | 1.85 |
| | R3GAN (Huang et al., 2025) | 21M | 1 | 1.96 |
| | CTM (Kim et al., 2024) | 59M | 1 | 1.98 |
| | SiD²A (Zhou et al., 2024) | 56M | 1 | 1.50 |
| Diffusion | DDPM (Ho et al., 2020) | 36M | 1000 | 3.17 |
| | iDDPM (Nichol & Dhariwal, 2021) | 50M | 4000 | 2.90 |
| | DDIM (Song et al., 2021a) | 36M | 100 | 4.16 |
| | DPM-Solver (Lu et al., 2022) | 36M | 48 | 2.65 |
| | DPM-Solver-v3 (Zheng et al., 2023) | 36M | 25 | 2.00 |
| | NCSN++ (Song et al., 2021b) | 108M | 2000 | 2.20 |
| | LSGM (Vahdat et al., 2021) | 376M | 138 | 2.10 |
| | EDM (Karras et al., 2022) | 56M | 35 | 1.97 |
| | EVODiff (Li et al., 2025) | 36M | 15 | 2.06 |
| Ours | DPM-Solver++ w/ DiFA | 36M | 15 | 2.02 |
| | DPM-Solver++ w/ DiFA | 36M | 20 | *1.96* |

*Table 2.* Results on class-conditional ImageNet-64. DiFA significantly outperforms standard diffusion baselines (e.g., ADM, iDDPM) with an order of magnitude fewer steps.

| Type | Model | #Params | NFE | FID ↓ |
|---|---|---|---|---|
| GAN† | StyleGAN-XL (Sauer et al., 2022) | 135M | 1 | 1.51 |
| | R3GAN (Huang et al., 2025) | 104M | 1 | 2.09 |
| | CTM (Kim et al., 2024) | 324M | 1 | 1.92 |
| | DMD2 (Yin et al., 2024) | 296M | 1 | 1.28 |
| | SiD²A (Zhou et al., 2024) | 296M | 1 | 1.11 |
| Diffusion | iDDPM (Nichol & Dhariwal, 2021) | 270M | 250 | 2.92 |
| | ADM (Dhariwal & Nichol, 2021) | 296M | 250 | 2.07 |
| | RIN (Jabri et al., 2022) | 281M | 1000 | 1.23 |
| | EDM (Karras et al., 2022) | 296M | 511 | 1.36 |
| | VDM++ (Kingma & Gao, 2023) | 296M | 511 | 1.43 |
| | DisCo-Diff (Xu et al., 2024b) | - | 623 | 1.22 |
| | EDM2-S (Karras et al., 2024b) | 280M | 63 | 1.58 |
| | DPM-Solver++ (Base) | 296M | 25 | 1.83 |
| | UniPC (Base) | 296M | 25 | 1.73 |
| Ours | DPM-Solver++ w/ DiFA | 296M | 25 | 1.64 |
| | UniPC w/ DiFA | 296M | 25 | *1.63* |

for the CFG-enabled systematic evaluation reported in Table 14, which is conducted on NVIDIA H200 GPUs. NFE denotes the number of neural network evaluations. Crucially, DiFA is implemented as a plug-and-play wrapper around existing solvers (DDIM, DPM-Solver++, UniPC, Heun) without any additional training or model fine-tuning. We use the same initial noise seeds for baselines and DiFA to ensure strict fairness. Code is available at DiFA.

### 5.2. Main Results

We now present a comprehensive evaluation demonstrating that DiFA consistently improves the evaluated training-free solvers across various regimes, effectively bridging the gap between rapid prototyping and high-fidelity generation.

**Surpassing the Efficiency Barrier on CIFAR-10.** As illustrated in the top row of Figure 6, the qualitative comparison in Figure 3, and the SOTA comparison in Table 1, DiFA

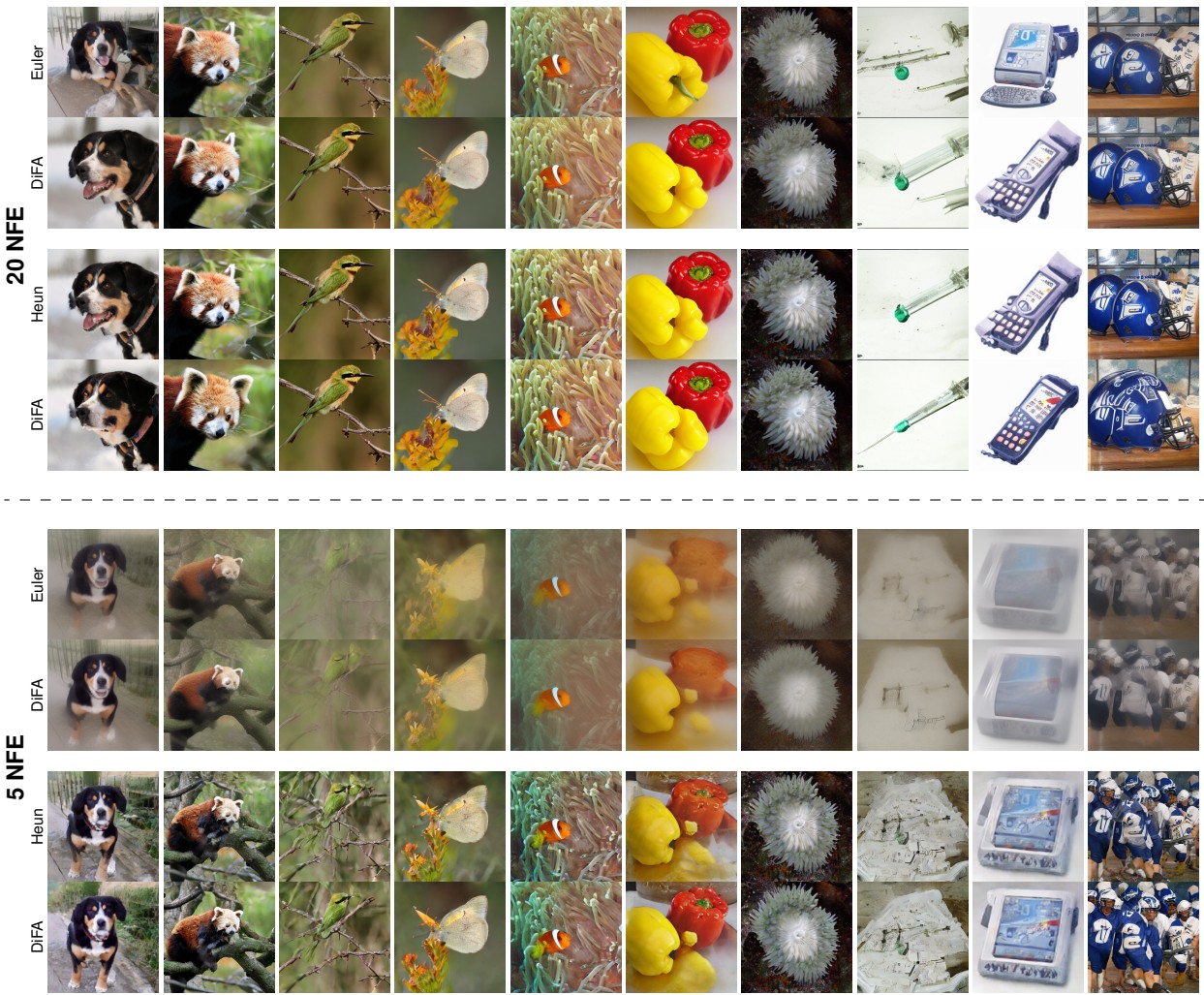

*Figure 4.* Visual quality comparison of generated samples on ImageNet 256 × 256 using the pre-trained SiT-XL/2 with a classifier-free guidance (CFG) scale of 1.5. SiT is a DiT-based architecture trained via Flow Matching. We compare the baseline Euler and Heun samplers against their corresponding refined predictions utilizing our proposed DiFA framework. The results demonstrate that DiFA significantly enhances the visual fidelity and generation quality of the baseline samplers.

substantially improves the trade-off between quality and computational cost. In the extremely low-budget regime (5–8 steps), where standard high-order solvers typically suffer from severe discretization errors due to unstable derivative estimation, DiFA effectively mitigates these issues via its temporal consensus mechanism. For instance, at NFE 8, DiFA reduces the FID of DPM-Solver++ from 8.40 to **4.15**, achieving a relative improvement of over **50%** and turning severely degraded few-step outputs into high-quality samples. Moving to the mid-NFE regime (10–15 steps), which is critical for real-world applications, DiFA exhibits clear dominance. At **NFE 12**, it achieves an FID of **2.18** (compared to the baseline's 3.70), surpassing the baseline by a large margin and outperforming the reported 12-step result of EVODiff (FID 2.25). Furthermore, even as the number of steps increases to 20 where baseline solvers typically

saturate, DiFA continues to yield gains (FID 1.96 vs. 2.33). This indicates that orthogonal deviation guidance helps preserve residual details that may otherwise be attenuated by consensus-based stabilization, allowing the model to break the FID 2.0 barrier without extra training.

**Universal Consistency on ImageNet.** The bottom row of Figure 6 and Table 2 highlight the robustness of DiFA on the more complex ImageNet manifold. The proposed framework shows broad solver compatibility: DiFA consistently lowers the FID curve when coupled with either the predictor-corrector-based UniPC or the high-order DPM-Solver++. Notably, DiFA substantially improves the unstable low-step behavior of the Heun solver, reducing FID from 230.05 to 110.20 at NFE 5. This suggests that the proposed prediction-alignment mechanism can mitigate severe trajectory drift in challenging low-NFE regimes. Moreover, DiFA proves

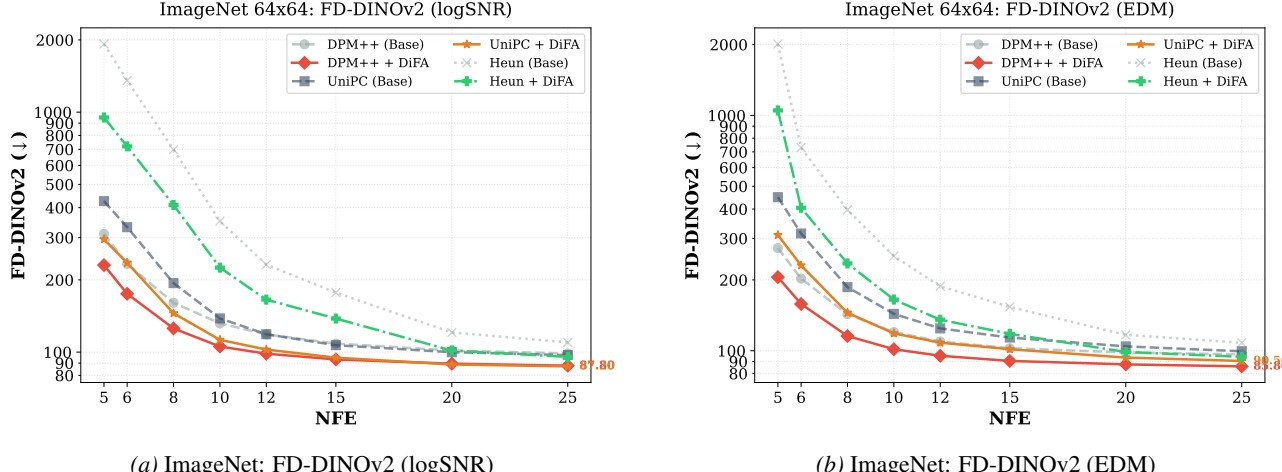

*(a)* ImageNet: FD-DINOv2 (logSNR)    *(b)* ImageNet: FD-DINOv2 (EDM)

*Figure 5.* Perceptual Quality (FD-DINOv2). DiFA also yields consistent improvements in FD-DINOv2 scores, indicating that our method enhances not just statistical fidelity (FID) but also perceptually aligned image quality.

effective under both logSNR and EDM noise schedules, suggesting that the proposed prediction-alignment principle is not tied to a specific discretization choice. At convergence (NFE 25), DiFA reaches an FID of **1.63–1.64** on ImageNet-64, showing strong training-free sampling performance among the evaluated solver configurations. This result suggests that refining clean-signal predictions can better exploit the capacity of pretrained diffusion models without additional network evaluations.

**Perceptual Quality Enhancement.** Beyond FID, FD-DINOv2 provides a perceptually aligned evaluation of image quality. Figure 5 demonstrates that DiFA yields consistent improvements in FD-DINOv2 scores. Unlike standard averaging methods that might improve FID at the cost of blurriness, DiFA achieves perceptual fidelity through a dual mechanism for structural filtering and texture enhancement, suggesting that the improvement is not obtained merely by sacrificing perceptual sharpness or semantic coherence.

To further examine whether DiFA is tied to pixel-space diffusion or a specific solver family, we provide further component, robustness, and cross-paradigm studies in Appendix B. On LSUN Bedroom, as shown in Table 4, DiFA improves a latent-diffusion baseline across 5, 10, and 20 NFE, demonstrating its effectiveness beyond the main pixel-space setting. Controlled EDM ablations show that historical consensus provides the main contribution, while SNR gating and magnitude alignment offer additional stabilization. Sensitivity analyses of the temporal-window, SNR-gating, residual-modulation, and logSNR-compatibility parameters further confirm robustness across different hyperparameter settings. Finally, experiments on SiT-XL/2 with ImageNet $256 \times 256$ show consistent FID and IS improvements under both unguided and CFG-enabled flow matching settings; qualitative examples are shown in Figure 4.

## Conclusion and Limitations

We presented DiFA, a training-free inference framework that reframes clean-signal prediction refinement as a sequential state-estimation problem. Rather than treating each denoising prediction as an exact estimate, DiFA exploits the common-anchor geometry and noise-level-dependent reliability structure induced by the forward process. It constructs a causal temporal consensus from historical predictions and applies deviation guidance to preserve informative residual details. Experiments across diffusion and flow-matching settings demonstrate that DiFA consistently improves sampling quality without additional network evaluations, supporting the effectiveness and broader applicability of prediction alignment. The current framework nevertheless relies on clean-prediction parameterizations, fixed hyperparameters, and an idealized static-anchor assumption. Future work will explore adaptive strategies, improved filtering mechanisms, and theoretical extensions for temporally correlated errors, as well as applications to text-to-image, video generation, inverse problems and distillation pipelines.

## Acknowledgements

This work was supported in part by grants from National Natural Science Foundation of China (52539005), the China Scholarship Council (202306150167), the fundamental research program of Guangdong, China (2023A1515011281), Guangdong Basic and Applied Basic Research Foundation (24202107190000687), Foshan Science and Technology Research Project (2220001018608).

## Impact Statement

This work introduces a training-free inference-time paradigm for improving generative-model sampling by refining clean-signal predictions rather than retraining models or replacing numerical solvers. By exploiting temporal prediction redundancy, DiFA can reduce the computational cost of high-quality generation and may make strong diffusion and flow-matching models more accessible under limited computational budgets. At the same time, more efficient generation may also lower the barrier to misuse, including the creation of misleading or harmful synthetic content. Since DiFA does not introduce new training data or modify model weights, these risks are largely inherited from the underlying generative models. We therefore encourage DiFA to be used together with existing safety filters, provenance mechanisms, and responsible deployment practices.

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

# Appendix

## A. Proofs

### A.1. Proof of Proposition 4.2

*Proof.* Under the ideal forward-aligned observation model, each observation is given by

$$y_i = x_{0,\star} + \eta_i, \qquad \mathrm{Cov}(\eta_i) = R_i I, \qquad R_i = \frac{c}{\mathrm{SNR}(t_i)}. \tag{23}$$

We consider a linear unbiased estimator

$$\hat{x}_{0,\mathrm{ideal}} = \sum_{i=1}^{n} w_i y_i, \qquad \sum_{i=1}^{n} w_i = 1. \tag{24}$$

Because the observation errors are mutually uncorrelated, the covariance of the estimator is

$$\mathrm{Cov}\left(\hat{x}_{0,\mathrm{ideal}}\right) = \sum_{i=1}^{n} w_i^2 R_i I. \tag{25}$$

Minimizing the scalar coefficient $\sum_{i=1}^{n} w_i^2 R_i$ subject to $\sum_{i=1}^{n} w_i = 1$ gives the Lagrangian

$$\mathrm{L} = \sum_{i=1}^{n} w_i^2 R_i - \lambda \left( \sum_{i=1}^{n} w_i - 1 \right). \tag{26}$$

Taking derivatives with respect to $w_i$ yields

$$\frac{\partial \mathrm{L}}{\partial w_i} = 2 w_i R_i - \lambda = 0, \tag{27}$$

and therefore

$$w_i = \frac{R_i^{-1}}{\sum_{j=1}^{n} R_j^{-1}} = \frac{\mathrm{SNR}(t_i)}{\sum_{j=1}^{n} \mathrm{SNR}(t_j)}. \tag{28}$$

Substituting these weights into Eq. (25) gives

$$\mathrm{Cov}\left(\hat{x}_{0,\mathrm{ideal}}^{\star}\right) = \left( \sum_{i=1}^{n} R_i^{-1} \right)^{-1} I = \frac{c}{\sum_{i=1}^{n} \mathrm{SNR}(t_i)} I. \tag{29}$$

Since $\sum_{i=1}^{n} R_i^{-1} > R_j^{-1}$ for every $j$ whenever $n \geq 2$, it follows that

$$\mathrm{Cov}\left(\hat{x}_{0,\mathrm{ideal}}^{\star}\right) \prec R_j I, \qquad j = 1, \ldots, n. \tag{30}$$

Thus, the ideal precision-weighted fused estimator has strictly lower covariance than any individual observation.

This result characterizes the ideal forward-aligned observation model. In DiFA, practical denoiser predictions are organized according to the resulting anchor-consistency and reliability-ordering principles, rather than being assumed to satisfy the same covariance model exactly.

The proof is complete. □

### A.2. Proof of Theorem 4.3

*Proof.* For the static anchor model, let $\hat{x}_{0,i}^{\mathrm{rec}}$ denote the recursive estimate after processing the first $i$ observations, and let $p_i I$ denote its estimation covariance. We initialize the recursion using the first observation:

$$\hat{x}_{0,1}^{\mathrm{rec}} = y_1, \qquad p_1 = R_1. \tag{31}$$

For $i = 2, \ldots, n$, the static-state Kalman gain is

$$k_i = \frac{p_{i-1}}{p_{i-1} + R_i}, \tag{32}$$

and the recursive update is

$$\hat{\boldsymbol{x}}_{0,i}^{\mathrm{rec}} = \hat{\boldsymbol{x}}_{0,i-1}^{\mathrm{rec}} + k_i \left( \boldsymbol{y}_i - \hat{\boldsymbol{x}}_{0,i-1}^{\mathrm{rec}} \right). \tag{33}$$

The corresponding covariance update is

$$p_i = (1 - k_i)p_{i-1} = \frac{p_{i-1}R_i}{p_{i-1} + R_i}, \tag{34}$$

which implies

$$p_i^{-1} = p_{i-1}^{-1} + R_i^{-1}. \tag{35}$$

Since $p_1 = R_1$, repeated application yields

$$p_n^{-1} = \sum_{i=1}^{n} R_i^{-1}. \tag{36}$$

Next, the recursive mean update can be rewritten as

$$\hat{\boldsymbol{x}}_{0,i}^{\mathrm{rec}} = \frac{R_i \hat{\boldsymbol{x}}_{0,i-1}^{\mathrm{rec}} + p_{i-1} \boldsymbol{y}_i}{p_{i-1} + R_i}. \tag{37}$$

Combining this relation with the covariance update gives

$$p_i^{-1} \hat{\boldsymbol{x}}_{0,i}^{\mathrm{rec}} = p_{i-1}^{-1} \hat{\boldsymbol{x}}_{0,i-1}^{\mathrm{rec}} + R_i^{-1} \boldsymbol{y}_i. \tag{38}$$

Using

$$p_1^{-1} \hat{\boldsymbol{x}}_{0,1}^{\mathrm{rec}} = R_1^{-1} \boldsymbol{y}_1, \tag{39}$$

we obtain

$$p_n^{-1} \hat{\boldsymbol{x}}_{0,n}^{\mathrm{rec}} = \sum_{i=1}^{n} R_i^{-1} \boldsymbol{y}_i. \tag{40}$$

Substituting Eq. (36) into Eq. (40) yields

$$\hat{\boldsymbol{x}}_{0,n}^{\mathrm{rec}} = \frac{\sum_{i=1}^{n} R_i^{-1} \boldsymbol{y}_i}{\sum_{i=1}^{n} R_i^{-1}}. \tag{41}$$

Finally, since

$$R_i^{-1} = \frac{\mathrm{SNR}(t_i)}{c}, \tag{42}$$

the common scale factor $c$ cancels out, giving

$$\hat{\boldsymbol{x}}_{0,n}^{\mathrm{rec}} = \frac{\sum_{i=1}^{n} \mathrm{SNR}(t_i) \boldsymbol{y}_i}{\sum_{i=1}^{n} \mathrm{SNR}(t_i)} = \hat{\boldsymbol{x}}_{0,\mathrm{ideal}}^{\star}. \tag{43}$$

Therefore, under the ideal independent-view observation model, the full-history precision-weighted anchor estimator is exactly equivalent to its static-state Kalman recursive realization.

The proof is complete. $\qquad\qquad\square$

### A.3. Lightweight Instantiation Used in Experiments

This section specifies the lightweight DiFA instantiation used in the main pixel-space diffusion experiments. The main text formulates DiFA through a causal historical reference anchor and an anchor-relative deviation modulation operator. Here, we provide the explicit forms of the alignment, compatibility weighting, and deviation guidance modules, as well as the default hyperparameter settings used to obtain the main experimental results. Additional configurations used for latent-diffusion and flow-matching validation are specified separately in Appendix B.

**Causal History Buffer.** At reverse step $t_i$, the history buffer stores at most $K$ previously processed clean-signal predictions and their corresponding logSNR values:

$$\mathrm{H}_K(t_i) = \left\{ \left( \hat{\boldsymbol{x}}_0^{(t_j)}, \ell_j \right) \right\}_{t_j \in \mathrm{W}_K(t_i)}, \qquad \ell_j = \log \mathrm{SNR}(t_j), \tag{44}$$

where $\mathrm{W}_K(t_i) \subseteq \{t_j : j > i\}$ contains the most recent historical steps along the reverse trajectory. The instantaneous prediction $\hat{\boldsymbol{x}}_0^{(t_i)}$ is used only as a query for constructing the historical reference and is not included among the aggregated reference values.

**Historical Prediction Alignment.** To reduce local scale mismatch between the current prediction and historical candidates, we apply channel-wise affine mean–variance alignment. Let $\mu_c(\boldsymbol{x})$ and $\sigma_c(\boldsymbol{x})$ denote the spatial mean and standard deviation of channel $c$. The aligned historical prediction is defined by

$$\tilde{\boldsymbol{x}}_{0,c}^{(t_j)} = \frac{\hat{\boldsymbol{x}}_{0,c}^{(t_j)} - \mu_c\left( \hat{\boldsymbol{x}}_0^{(t_j)} \right)}{\sigma_c\left( \hat{\boldsymbol{x}}_0^{(t_j)} \right) + \epsilon_{\mathrm{a}}} \cdot \sigma_c\left( \hat{\boldsymbol{x}}_0^{(t_i)} \right) + \mu_c\left( \hat{\boldsymbol{x}}_0^{(t_i)} \right), \tag{45}$$

where $\epsilon_{\mathrm{a}} > 0$ is a numerical stabilizer. This operation matches the channel-wise first- and second-order statistics of each historical prediction to those of the current prediction before reference aggregation.

**Structural and Noise-Level Compatibility.** We compute structural compatibility after lightweight spatial average pooling. Let $\mathrm{P}_{\mathrm{sim}}(\cdot)$ denote average pooling with kernel size $k_{\mathrm{sim}} \times k_{\mathrm{sim}}$. For each historical candidate, the local similarity score is

$$m_{ij} = \frac{\left\langle \mathrm{vec}\left( \mathrm{P}_{\mathrm{sim}}\left( \hat{\boldsymbol{x}}_0^{(t_i)} \right) \right), \mathrm{vec}\left( \mathrm{P}_{\mathrm{sim}}\left( \tilde{\boldsymbol{x}}_0^{(t_j)} \right) \right) \right\rangle}{\left\| \mathrm{vec}\left( \mathrm{P}_{\mathrm{sim}}\left( \hat{\boldsymbol{x}}_0^{(t_i)} \right) \right) \right\|_2 \left\| \mathrm{vec}\left( \mathrm{P}_{\mathrm{sim}}\left( \tilde{\boldsymbol{x}}_0^{(t_j)} \right) \right) \right\|_2 + \epsilon_{\mathrm{s}}}, \tag{46}$$

where $\epsilon_{\mathrm{s}} > 0$ prevents numerical instability. The similarity scores are normalized within the current history window:

$$\bar{m}_{ij} = \frac{m_{ij} - \mu_{m,i}}{\sigma_{m,i} + \epsilon_{\mathrm{n}}}, \qquad \mu_{m,i} = \frac{1}{|\mathrm{W}_K(t_i)|} \sum_{t_j \in \mathrm{W}_K(t_i)} m_{ij}, \tag{47}$$

where $\sigma_{m,i}$ is the standard deviation of the similarity scores within the current temporal window. The compatibility logit combines structural agreement and logSNR proximity:

$$q_{ij} = \tau \bar{m}_{ij} - \mu(\ell_i) |\ell_i - \ell_j|, \tag{48}$$

where $\tau$ controls the sharpness of structural selection and $\mu(\ell_i)$ controls the strength of noise-level compatibility at the current reverse step. The normalized historical weights are

$$a_{ij} = \frac{\exp(q_{ij})}{\sum_{t_k \in \mathrm{W}_K(t_i)} \exp(q_{ik})}. \tag{49}$$

The causal historical reference anchor is then

$$\hat{\boldsymbol{x}}_0^{\mathrm{cons}}(t_i) = \sum_{t_j \in \mathrm{W}_K(t_i)} a_{ij} \tilde{\boldsymbol{x}}_0^{(t_j)}. \tag{50}$$

**Deviation Modulation.** Given the historical reference anchor, we compute the anchor-relative deviation:

$$\boldsymbol{r}_{t_i} = \hat{\boldsymbol{x}}_0^{(t_i)} - \hat{\boldsymbol{x}}_0^{\mathrm{cons}}(t_i). \tag{51}$$

To reduce residual corrections dominated by magnitude variation parallel to the current prediction, we apply an orthogonal residual projection:

$$\boldsymbol{r}_{t_i}^{\perp} = \boldsymbol{r}_{t_i} - \frac{\left\langle \boldsymbol{r}_{t_i}, \hat{\boldsymbol{x}}_0^{(t_i)} \right\rangle}{\left\| \hat{\boldsymbol{x}}_0^{(t_i)} \right\|_2^2 + \epsilon_{\mathrm{p}}} \hat{\boldsymbol{x}}_0^{(t_i)}, \tag{52}$$

where $\epsilon_{\mathrm{p}} > 0$ is a numerical stabilizer. We then decompose the projected residual into locally smoothed and high-frequency components:

$$\boldsymbol{r}_{t_i}^{\mathrm{low}} = \mathrm{P}_{\mathrm{res}}\left(\boldsymbol{r}_{t_i}^{\perp}\right), \qquad \boldsymbol{r}_{t_i}^{\mathrm{high}} = \boldsymbol{r}_{t_i}^{\perp} - \boldsymbol{r}_{t_i}^{\mathrm{low}}, \tag{53}$$

where $\mathrm{P}_{\mathrm{res}}(\cdot)$ denotes local average pooling with kernel size $k_{\mathrm{res}} \times k_{\mathrm{res}}$. The high-frequency gate is defined as

$$\lambda_{\mathrm{hf}}(\ell_i) = \frac{1}{1 + \exp(-\ell_i)}. \tag{54}$$

The deviation modulation operator used in our experiments is therefore instantiated as

$$\mathrm{G}\left(\boldsymbol{r}_{t_i}, \hat{\boldsymbol{x}}_0^{(t_i)}, \ell_i\right) = \boldsymbol{r}_{t_i}^{\mathrm{low}} + \lambda_{\mathrm{hf}}(\ell_i)\boldsymbol{r}_{t_i}^{\mathrm{high}}. \tag{55}$$

Finally, the refined clean-signal prediction is

$$\hat{\boldsymbol{x}}_0^{\mathrm{DiFA}}(t_i) = \hat{\boldsymbol{x}}_0^{(t_i)} + \omega\, \mathrm{G}\left(\boldsymbol{r}_{t_i}, \hat{\boldsymbol{x}}_0^{(t_i)}, \ell_i\right). \tag{56}$$

**Default Hyperparameters.** Unless otherwise stated, all reported DiFA results use the lightweight instantiation above with the default settings in Table 3. Ablation experiments modify only the factors explicitly specified in their corresponding tables.

*Table 3.* Default hyperparameter settings for the lightweight DiFA instantiation used in the main experiments.

| Hyperparameter | Meaning | Default value |
|---|---|---|
| $K$ | causal temporal window size | 3 |
| $\tau$ | structural compatibility sharpness | 4.0 |
| $s$ | reported DiFA deviation scale | 1.7 |
| $\omega = s - 1$ | residual guidance coefficient | 0.7 |
| $k_{\mathrm{sim}}$ | similarity pooling kernel size | 3 |
| $k_{\mathrm{res}}$ | residual smoothing kernel size | 5 |
| $\lambda_{\mathrm{hf}}(\ell_i)$ | high-frequency gate | $(1 + \exp(-\ell_i))^{-1}$ |

The reported scale $s$ follows the convention that $s = 1$ recovers the uncorrected baseline; therefore, the coefficient multiplying the deviation guidance is implemented as $\omega = s - 1$. The function $\mu(\ell_i)$ and the numerical stabilizers $\epsilon_{\mathrm{a}}$, $\epsilon_{\mathrm{s}}$, $\epsilon_{\mathrm{n}}$, and $\epsilon_{\mathrm{p}}$ are set according to the implementation configuration used for each experimental setting.

## B. Additional Ablation and Validation Studies

This appendix provides additional ablation, sensitivity, and validation studies that complement the main experiments and further clarify the contribution of each component in DiFA. Unless otherwise specified, all ablation results are obtained under the same pretrained model and solver setting as the corresponding baseline. Lower FID is better, and higher IS is better.

### B.1. Ablation on Latent Diffusion Models

We first evaluate DiFA on a latent diffusion model using the LSUN Bedroom dataset. For this experiment, we retain the notation used in the corresponding implementation, where $\gamma_0$ and $\lambda$ denote the deviation strength and residual coefficient, respectively. The latent-diffusion experiments use a task-specific parameterization; the corresponding symbols are defined in Table 6. This experiment verifies that DiFA is not restricted to pixel-space diffusion models. Since DiFA operates on clean-signal predictions within the sampling loop, it can also be applied to latent-space samplers without changing the pretrained network.

The LDM ablation shows that historical consensus is essential: when the history window collapses to $W = 1$, DiFA degenerates to the baseline. SNR gating is particularly useful at medium and high NFE budgets, where unguided activation may introduce unstable corrections. The magnitude alignment factor $\phi$ stabilizes latent-space energy and prevents excessive magnitude drift during deviation guidance.

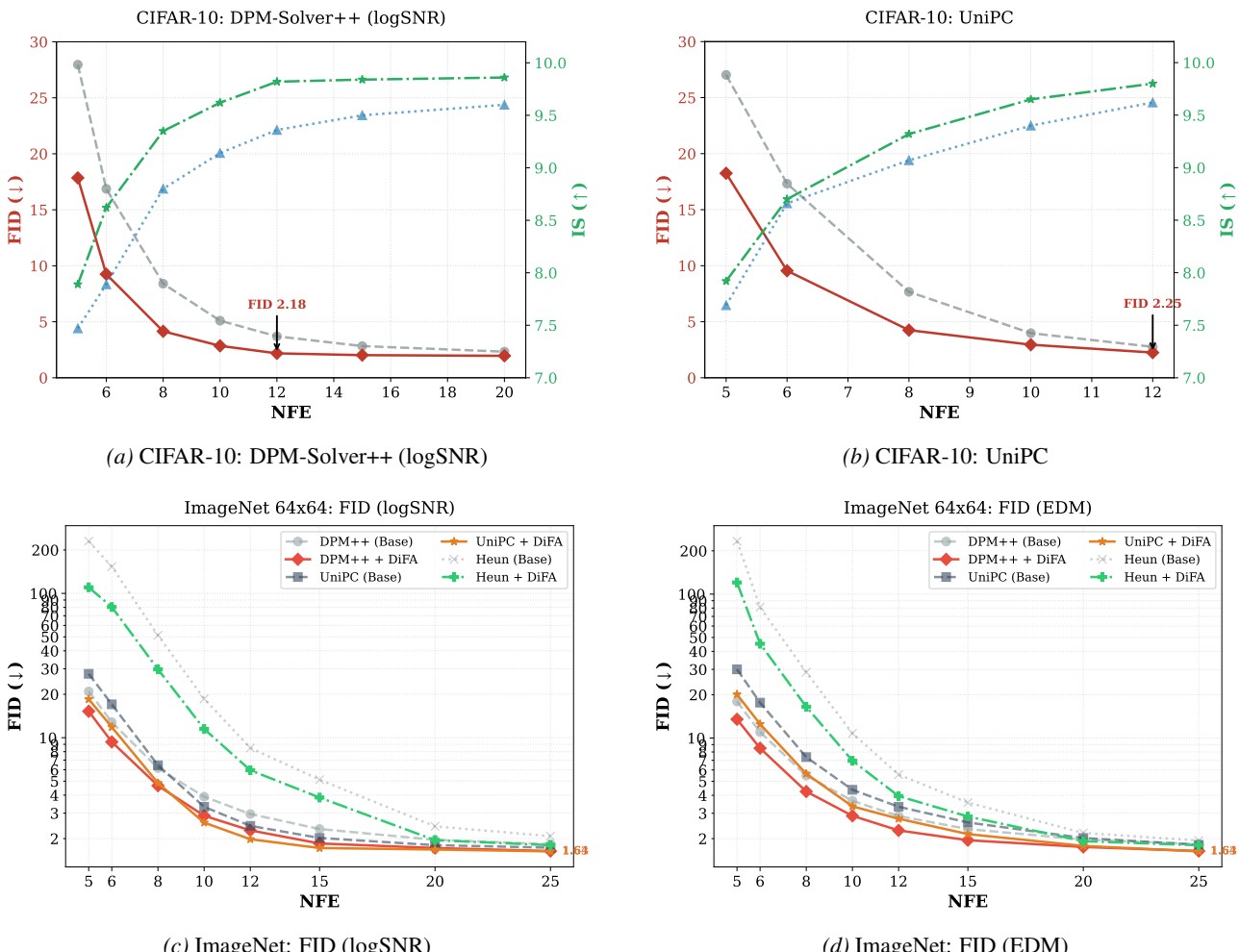

*Figure 6.* Quantitative Results. **Top Row:** On CIFAR-10, DiFA (solid red lines) significantly lowers FID compared to baselines (dashed gray lines) across all NFE regimes for both DPM-Solver++ and UniPC. Note the rapid convergence at NFE 12 (FID ≈ 2.2). **Bottom Row:** On ImageNet 64x64, DiFA consistently improves the convergence of various solvers (DPM++, UniPC, Heun) under both logSNR and EDM schedules, achieving strong training-free performance with an FID of ≈ 1.63 at 25 steps.

The robustness study shows that DiFA consistently improves over the naive baseline under different hyperparameter choices. The best configuration can vary across NFE budgets, suggesting that DiFA is not overly sensitive to a single parameter setting.

## B.2. Ablation on EDM

We further conduct controlled ablation experiments under the EDM setting at 10 NFE. These results isolate the contribution of the main DiFA components: historical consensus, adaptive SNR gating, and magnitude alignment.

The main component ablation under the EDM setting is reported in Table 7. The full DiFA configuration improves the baseline FID from 5.071 to 2.797, as reported in Table 7. The corresponding qualitative comparison is shown in Figure 3. The identical results of the naive baseline and the w/o History setting indicate that setting $W = 1$ disables temporal consensus and collapses the update to the baseline behavior.

The following sensitivity studies examine the robustness of DiFA to individual hyperparameters under the EDM setting. Each sweep is designed to vary one hyperparameter while keeping the remaining settings fixed within that sweep. Since the controlled configurations may differ across sweeps, the results should be compared primarily within each table rather than across tables.

*Table 4.* DiFA versus the naive ODE-solver baseline on LSUN Bedroom. The naive baseline corresponds to DPM-Solver++ with DiFA disabled.

| Method | 5 NFE | 10 NFE | 20 NFE |
|---|---|---|---|
| Naive Baseline | 21.238 | 3.877 | 3.256 |
| DiFA (Full) | **5.912** | **3.579** | **2.939** |
| FID Improvement | 72.2% | 7.7% | 9.7% |

*Table 5.* Component ablation on LSUN Bedroom. We isolate the effects of historical consensus, SNR gating, and magnitude alignment.

| Configuration | $W$ | $SNR_{\mathrm{lo}}$ | $\phi$ | 5 NFE $\downarrow$ | 10 NFE $\downarrow$ | 20 NFE $\downarrow$ |
|---|---|---|---|---|---|---|
| Naive Baseline | 1 | Full | 0.0 | 21.238 | 3.877 | 3.256 |
| w/o History | 1 | -4.5 | 0.5 | 21.238 | 3.877 | 3.256 |
| w/o SNR Gating | 3 | Full | 0.5 | **5.838** | 4.414 | 3.119 |
| w/o Energy ($\phi$) | 3 | -4.5 | 0.0 | 7.053 | 4.462 | 3.142 |
| DiFA (Full) | 3 | -4.5 | 0.5 | 5.912 | **3.579** | **2.939** |

The performance exhibits a U-shaped trend with respect to $W$. A very small window fails to exploit temporal redundancy, while an overly large window introduces stale predictions and causes historical lag. We therefore use $W = 3$ as the default setting.

The SNR gating threshold controls when DiFA starts to apply alignment. Early activation may inject corrections during noisy and unstable stages, while overly late activation may miss useful structural formation stages. Within this sweep, moderate activation gives the best FID, and the IS values remain stable across nearby thresholds.

The factor $\phi$ stabilizes the magnitude of the deviation-guided prediction. A moderate value preserves residual details while avoiding excessive latent-energy drift.

The deviation scale $\gamma_0$ controls the strength of detail re-injection. Too small a value under-corrects the consensus-smoothed prediction, whereas too large a value may amplify unstable residuals. In this sweep, $\gamma_0 = 1.4$ gives the lowest FID, while $\gamma_0 = 1.5$ gives the highest IS.

The logSNR compatibility coefficient $\mu$ controls how strongly DiFA penalizes temporally distant or noise-incompatible historical predictions. A small compatibility coefficient is generally stable. In this sweep, $\mu = 0.2$ gives the lowest FID, while $\mu = 0.1$ gives the highest IS, suggesting that values in the range of 0.1–0.3 provide a reliable balance between using historical evidence and avoiding stale predictions.

### B.3. Cross-Paradigm Validation on Flow-Matching Models

Although DiFA is introduced in the context of diffusion sampling, its core operation is not tied to a specific diffusion solver. DiFA refines the clean-signal prediction supplied to a downstream update rule. Therefore, it can also be applied to flow-matching models when the predicted velocity or transport state can be converted into an endpoint or clean-signal estimate. This setting provides a useful test of whether the proposed prediction-alignment principle generalizes beyond conventional diffusion samplers.

We evaluate this transferability in two complementary settings. The first is an unguided validation on SiT-XL/2 with ImageNet (256×256), where classifier-free guidance (CFG) is disabled. This experiment was conducted under a lightweight computational setting on 4090 GPUs and was designed as a sanity check for cross-paradigm transfer. As shown in Table 13, DiFA improves the ODE-solver baseline at 5, 10, and 50 NFE without classifier-free guidance. These results indicate that the temporal redundancy exploited by DiFA is also present in flow-matching trajectories.

The second setting provides a more systematic validation after further adapting DiFA to the flow-matching formulation. In this FM-adapted version, DiFA is applied to the endpoint prediction recovered from the flow-matching transport trajectory, and the temporal consensus is computed in a clean-signal space rather than directly on the velocity field. We also use the flow-matching time variable to define a logSNR-like reliability coordinate, so that the sliding-window consensus and deviation guidance remain compatible with the FM trajectory. This adaptation makes DiFA a clean-prediction refinement

*Table 6.* Hyperparameter robustness on LSUN Bedroom. A dash denotes the naive baseline without DiFA. All listed DiFA configurations improve over the naive baseline.

| $\gamma_0$ | $\lambda$ | $\phi$ | $SNR_{\text{lo}}$ | $SNR_{\text{hi}}$ | $\tau$ | 5 NFE $\downarrow$ | 10 NFE $\downarrow$ | 20 NFE $\downarrow$ |
|---|---|---|---|---|---|---|---|---|
| – | – | – | – | – | – | 21.238 | 3.877 | 3.256 |
| 1.40 | 0.5 | 0.5 | -2.5 | 3.0 | 0.8 | 9.142 | 3.684 | 3.230 |
| 1.40 | 0.5 | 0.5 | -2.5 | 4.0 | 0.8 | 9.032 | 3.729 | 3.248 |
| 1.50 | 0.5 | 0.5 | -4.5 | 3.0 | 0.8 | 6.535 | 3.658 | 3.104 |
| 1.50 | 0.5 | 0.7 | -3.5 | 2.0 | 0.8 | 7.576 | **3.579** | 3.120 |
| 1.75 | 0.4 | 0.5 | -4.5 | 5.0 | 0.8 | 6.069 | 4.465 | **2.939** |
| 1.75 | 0.4 | 0.5 | -4.5 | 5.0 | 1.0 | 6.019 | 4.429 | 2.954 |

*Table 7.* Main component ablation under the EDM setting at 10 NFE.

| Configuration | History ($W$) | SNR Gating ($SNR_{\text{lo}}$) | Energy ($\phi$) | FID $\downarrow$ | IS $\uparrow$ |
|---|---|---|---|---|---|
| Naive Baseline | 1 | Full | 0.0 | 5.071 | 9.139 |
| w/o History | 1 | -2.5 | 0.5 | 5.071 | 9.139 |
| w/o SNR Gating | 3 | Full | 0.5 | 3.375 | 9.574 |
| w/o Energy ($\phi$) | 3 | -2.5 | 0.0 | 2.846 | 9.661 |
| DiFA (Full) | 3 | -2.5 | 0.5 | **2.797** | **9.680** |

module for flow-matching inference rather than a diffusion-specific correction.

Using this FM-adapted implementation, we conduct a CFG-enabled systematic evaluation on H200 GPUs with CFG (=1.5). Table 14 expands the preliminary validation into a full solver–NFE–scale matrix under both Euler and Heun2 solvers. This experiment is not intended as a component ablation. Instead, it evaluates cross-paradigm robustness: whether DiFA remains effective under stronger guided generation settings and whether its behavior is stable across solver choices, NFE budgets, and guidance strengths.

The two settings should therefore be interpreted as complementary evidence rather than directly comparable benchmarks. The preliminary no-CFG experiment verifies the basic transferability of DiFA to flow-matching models. The CFG-enabled H200 experiment, obtained after the FM-specific adaptation, provides a stronger and more realistic validation under guided sampling and multiple solver configurations.

The results in Table 14 show that DiFA consistently improves FID, IS, sFID, and precision over the corresponding flow-matching baselines. Recall remains comparable in most settings, although a small decrease is observed for Heun2 at 20 NFE, indicating a mild precision–recall trade-off in this regime. The gains are especially pronounced in the low-NFE regime, where trajectory discretization error and clean-signal prediction uncertainty are more severe. For example, in the CFG-enabled setting, DiFA reduces the Euler 5-NFE FID from (52.64) to (27.61), and reduces the Heun2 5-NFE FID from (14.64) to (7.99). At 10 NFE, DiFA still provides substantial improvements for both Euler and Heun2. At 20 NFE, where the baseline trajectories are already stronger, the relative margin naturally becomes smaller but remains generally positive.

The DiFA refinement-scale sweep further shows that the improvement is not caused by a single isolated hyperparameter choice. Scales around (1.7)–(1.75) usually provide the strongest FID and IS performance, while smaller scales still improve over the corresponding baselines. These results support the view that DiFA acts as a solver-compatible inference-time refinement mechanism and that its temporal prediction-alignment principle can extend from diffusion samplers to flow-matching trajectories.

The quantitative improvements are also reflected in the qualitative comparison shown in Figure 7. Under the 5-NFE setting, DiFA produces samples with better structural coherence and fewer truncation artifacts than the baseline solver. The comparison provides qualitative evidence that the numerical gains are accompanied by improved visual quality.

*Table 8.* Sensitivity of the history window size $W$ under the EDM setting.

| Window Size ($W$) | 1 (Baseline) | 2 | 3 (Ours) | 4 |
|---|---|---|---|---|
| FID ↓ | 5.071 | 3.219 | **2.797** | 3.791 |
| IS ↑ | 9.139 | 9.443 | **9.680** | 9.593 |

*Table 9.* Sensitivity of the adaptive SNR gating threshold $SNR_{\text{lo}}$ under the EDM setting.

| SNR Threshold | -6.0 (Early) | -4.0 | -2.5 | -1.0 (Late) |
|---|---|---|---|---|
| FID ↓ | 3.137 | 3.040 | **2.986** | 3.068 |
| IS ↑ | 9.554 | **9.563** | 9.555 | 9.509 |

*Table 10.* Sensitivity of the magnitude alignment factor $\phi$ under the EDM setting.

| Rescale Factor ($\phi$) | 0.0 | 0.3 | 0.5 (Ours) | 0.8 | 1.0 |
|---|---|---|---|---|---|
| FID ↓ | 2.846 | 3.077 | **2.797** | 3.074 | 3.085 |
| IS ↑ | 9.661 | 9.555 | **9.680** | 9.568 | 9.565 |

*Table 11.* Sensitivity of the deviation boosting scale $\gamma_0$ under the EDM setting.

| Scale ($\gamma_0$) | 1.1 | 1.2 | 1.3 | 1.4 | 1.5 |
|---|---|---|---|---|---|
| FID ↓ | 4.119 | 3.470 | 3.121 | **3.071** | 3.285 |
| IS ↑ | 9.282 | 9.383 | 9.494 | 9.565 | **9.602** |

*Table 12.* Sensitivity of the compatibility coefficient $\mu$ under the EDM setting.

| LogSNR Compatibility Coefficient($\mu$) | 0.1 | 0.2 | 0.3 | 0.4 | 0.6 |
|---|---|---|---|---|---|
| FID ↓ | 3.212 | **3.205** | 3.220 | 3.256 | 3.373 |
| IS ↑ | **9.499** | 9.492 | 9.478 | 9.457 | 9.425 |

*Table 13.* Preliminary no-CFG validation on SiT-XL/2 with ImageNet $256 \times 256$. This lightweight ODE-solver experiment was conducted on 4090 GPUs to test whether DiFA transfers to flow-matching trajectories without classifier-free guidance.

| NFE | Method | IS ↑ | FID ↓ | sFID ↓ | Precision ↑ | Recall ↑ |
|---|---|---|---|---|---|---|
| 5 | ODE-Solver | 22.5016 | 94.5311 | 60.2238 | 0.2089 | 0.5016 |
| 5 | DiFA (Ours) | **31.8312** | **72.3969** | **40.0246** | **0.2905** | **0.5694** |
| 10 | ODE-Solver | 80.2694 | 27.8142 | 13.0268 | 0.5542 | 0.6091 |
| 10 | DiFA (Ours) | **91.7977** | **21.5990** | **10.2921** | **0.5982** | **0.6256** |
| 50 | ODE-Solver | 118.6470 | 11.1215 | 7.0554 | 0.6662 | 0.6682 |
| 50 | DiFA (Ours) | **120.2284** | **10.7515** | **6.8273** | **0.6676** | **0.6694** |

*Table 14.* CFG-enabled systematic validation of the FM-adapted DiFA on SiT-XL/2 with ImageNet 256 × 256. This experiment is conducted on NVIDIA H200 GPUs with CFG fixed at 1.5. We evaluate Euler and Heun2 solvers across multiple NFE budgets and DiFA refinement scales. Best and second-best results among the evaluated DiFA configurations are shown in bold and underlined, respectively.

| NFE | Method (Scale) | IS ↑ | (△ IS) | FID ↓ | (△ FID) | sFID ↓ | Prec. ↑ | Rec. ↑ |
|---|---|---|---|---|---|---|---|---|
| 5 | Baseline (Euler) | 58.02 | - | 52.64 | - | 44.10 | 0.3692 | 0.4353 |
|  | DiFA (1.25) | 73.90 | (+27.4%) | 40.81 | (-22.5%) | 33.33 | 0.4407 | 0.4290 |
|  | DiFA (1.5) | 86.49 | (+49.1%) | 32.86 | (-37.6%) | 26.11 | 0.4969 | 0.4322 |
|  | DiFA (1.7) | 94.15 | (+62.3%) | 28.52 | (-45.8%) | 22.23 | 0.5318 | 0.4383 |
|  | DiFA (1.75) | **95.88** | **(+65.2%)** | **27.61** | **(-47.5%)** | **21.46** | **0.5400** | **0.4406** |
| 10 | Baseline (Euler) | 184.11 | - | 8.35 | - | 8.83 | 0.7303 | 0.5234 |
|  | DiFA (1.25) | 200.94 | (+9.1%) | 6.08 | (-27.2%) | 6.40 | 0.7616 | 0.5311 |
|  | DiFA (1.5) | 213.98 | (+16.2%) | 4.80 | (-42.5%) | 5.44 | 0.7790 | 0.5342 |
|  | DiFA (1.7) | 219.79 | (+19.4%) | 4.28 | (-48.7%) | **5.40** | 0.7884 | 0.5361 |
|  | DiFA (1.75) | **220.91** | **(+20.0%)** | **4.20** | **(-49.7%)** | 5.47 | **0.7912** | **0.5378** |
| 20 | Baseline (Euler) | 231.60 | - | 3.33 | - | 5.33 | 0.7902 | 0.5736 |
|  | DiFA (1.25) | 241.02 | (+4.1%) | 2.76 | (-17.2%) | 4.45 | 0.7997 | 0.5770 |
|  | DiFA (1.5) | 248.54 | (+7.3%) | 2.48 | (-25.5%) | **4.24** | 0.8060 | 0.5784 |
|  | DiFA (1.7) | 251.81 | (+8.7%) | 2.42 | (-27.5%) | 4.42 | 0.8105 | 0.5778 |
|  | DiFA (1.75) | **252.71** | **(+9.1%)** | **2.41** | **(-27.5%)** | 4.51 | **0.8121** | **0.5804** |
| 5 | Baseline (Heun2) | 148.39 | - | 14.64 | - | 22.11 | 0.6590 | 0.5155 |
|  | DiFA (1.25) | 162.01 | (+9.2%) | 11.67 | (-20.3%) | 17.88 | 0.6827 | 0.5294 |
|  | DiFA (1.5) | 174.30 | (+17.5%) | 9.53 | (-34.9%) | 14.78 | 0.7049 | 0.5388 |
|  | DiFA (1.7) | 181.66 | (+22.4%) | 8.26 | (-43.6%) | 12.92 | 0.7158 | 0.5538 |
|  | DiFA (1.75) | **183.22** | **(+23.5%)** | **7.99** | **(-45.4%)** | **12.53** | **0.7178** | **0.5560** |
| 10 | Baseline (Heun2) | 233.24 | - | 3.15 | - | 6.59 | 0.7882 | 0.5852 |
|  | DiFA (1.25) | 241.89 | (+3.7%) | 2.66 | (-15.7%) | 5.65 | 0.7959 | 0.5885 |
|  | DiFA (1.5) | 247.92 | (+6.3%) | 2.37 | (-24.9%) | 5.09 | 0.7989 | 0.5894 |
|  | DiFA (1.7) | 253.58 | (+8.7%) | 2.22 | (-29.6%) | 4.86 | 0.8034 | **0.5934** |
|  | DiFA (1.75) | **254.45** | **(+9.1%)** | **2.19** | **(-30.3%)** | **4.82** | **0.8046** | 0.5925 |
| 20 | Baseline (Heun2) | 250.65 | - | 2.20 | - | 4.73 | 0.8009 | 0.6000 |
|  | DiFA (1.25) | 254.11 | (+1.4%) | 2.08 | (-5.6%) | 4.52 | 0.8039 | **0.5987** |
|  | DiFA (1.5) | 259.56 | (+3.6%) | 2.02 | (-8.4%) | **4.47** | 0.8076 | 0.5986 |
|  | DiFA (1.7) | 261.63 | (+4.4%) | **2.01** | **(-8.6%)** | 4.51 | 0.8103 | 0.5964 |
|  | DiFA (1.75) | **262.39** | **(+4.7%)** | 2.01 | (-8.6%) | 4.53 | **0.8117** | 0.5967 |

SiT with Baseline Solver         SiT with DiFA Refined Prediction

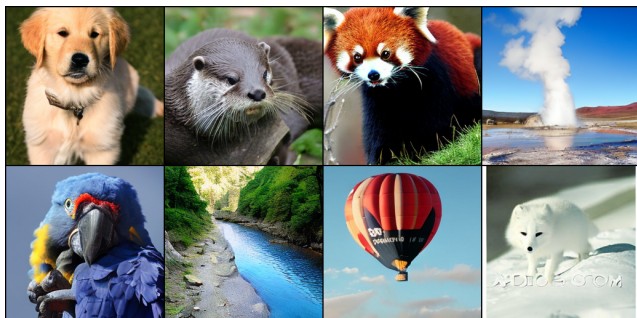

*Figure 7.* Qualitative comparison on ImageNet (256 × 256) using the pretrained SiT-XL/2 model with the Euler solver, 5 function evaluations (NFE = 5), and a classifier-free guidance (CFG) scale of 4.0. Compared with the baseline, DiFA preserves better structural coherence and local details under this aggressive few-step setting.

