# OpenReview forum: "DiFA: Inference-Time Forward-Process Alignment for Diffusion Models"
_ICML.cc/2026/Conference — ICML 2026 regular_

### Official Review · Reviewer_xfCq · 2026-03-04

**Soundness:** 4
**Presentation:** 3
**Significance:** 4
**Originality:** 4
**Overall Recommendation:** 5
**Confidence:** 4

**Summary:**

The paper argue that diffusion model degradation at lower number of steps should not only be attributed to the result of temporal discretization errors of the ODE solver but to also to the imperfect score estimation. The authors propose to reduce the variance of the latter by leveraging the full previous sample trajectories instead of only the current noisy state. This recasts diffusion inference as a sequential
state estimation problem that is optimally solved using Kalman filters. The resulting algorithm improves perceptual quality at inference time without re-training as long as it is not used in the later low-variance steps.

**Compliance With Llm Reviewing Policy:**

Affirmed.

**Key Questions For Authors:**

- Could the authors specify what the first-order property near line 209 refers to? I assume this mean 'sufficient' trajectory straightness?
- Which assumptions are needed for the mean and variance of the estimation error near eq. (7) to hold? This seems to ignore epistemic uncertainty, i.e. a badly (limited capacity, data) trained model surely does not have well-calibrated estimation errors?
- Is $\gamma_0$ and $\gamma(t)$ supposed to be related to the $\gamma$ defined earlier? If not, please rename for clarity.

**Limitations:**

No limitations discussed. Ablation without Deviation Boosting, and discussion of assumptions in eq. (7) could improve this.

**Strengths And Weaknesses:**

Strengths
- The paper provides an interesting non-standard perspective on diffusion model inference, arguing that not using the full previous sampling  trajectory is inefficient.
- The motivation behind their method is clear and has a strong provable theoretical basis.
- Experiments seem to strongly indicate the advantage of the proposed work.
- Clear and concise writing.

Weaknesses
- The uncorrelated case is theoretically sound but the correlated case relies on some key assumptions that should be empirically verified. The text mentions looking at correlation matrices but they are not in the paper. In fact, page 5 contains several broken links. Even then finding a proof for how eq. (12) solves eq. (9) under these correlation assumptions would make the paper much stronger.
- The paper is missing (at least in the appendix) a set of uncurated samples from the trained models.
- The motivation behind deviation boosting is mostly empirical. While I don't expect the authors to do this (or if it feasible) it would be interesting to see an analysis for $T \to \infty$ that investigates if this is required due to finite step sizes or a fundamental flaw of using the lowest-variance estimator. More practically speaking, an ablation of generation with deviation boosting (FID, uncurated samples) would strengthen the paper.
- Some notation is not properly defined ($\mathcal{I}$ in Proposition 4.2, $\mathcal{W}$ in eq.(12), etc.)
- The experimental setup should be described in more detail to ensure reproducibility. At a minimum, choices for $\gamma_0$ and $\lambda$ need to be discussed. Following from that it would be interesting to see which portion of the steps receives meaningful additional signal (i.e. when is $\gamma(t) \approx 1$).
- The baselines in Tables 1 and 2 feel slightly outdated and incomplete. As the paper provides several ablations of models with and without DiFA applied this is not a crucial flaw.
- Some minor details like a missing legend in Figure 3(a) and the MVUE abbreviation not being written out on its first occurrence should be fixed.

---

> ### Author Rebuttal · Authors · 2026-03-31
>
> We sincerely thank you for your meticulous and highly constructive feedback. Your expert insights have significantly strengthened the theoretical rigor of our paper. We have carefully addressed all your points below:
>
> > 1. Correlated Noise (W1) & Notation Clashes (W4, W7, Q3)
>
> We apologize for the broken links. We will formally introduce *Assumption B.1 (Exponential Correlation Decay)* in the Appendix. Setting our decay factor $\lambda \approx \rho$ effectively pre-whitens the observations before fusion, providing the theoretical bridge to Eq. (9).
> Additionally, to resolve the critical notation clash you identified, we will *rename the forgetting factor to $\mu$* throughout the paper, reserving $\gamma(t)$ strictly for the Deviation Boosting schedule. We will also explicitly define $\mathcal{W}(t)$, distinguish $\mathcal{I}$ notation, and fix the missing legends/acronyms.
>
> > 2. Epistemic Uncertainty & Assumptions (Q1, Q2)
>
> Your observation regarding Eq. (7) is sharp and completely correct. For a real, capacity-limited network, the epistemic bias means $\mathbb{E}[\eta_t|x_t] \neq 0$. While our fading memory mechanism ($\mu$) naturally mitigates this by aggressively discounting older, high-bias predictions, the underlying assumption remains a limitation. We will add a dedicated *Limitations section* explicitly discussing this epistemic uncertainty.
>
> Regarding the *'first-order property'*, we clarify that this refers to the drift field being constant along each generative trajectory, where the ODE trajectories form straight lines in data space with zero transport curvature. Under this geometric condition, the integrand does not change along the path, so first-order Euler integration incurs zero truncation error for a perfect model. In practice, however, two independent sources of error prevent this ideal from being realised: (1) standard pre-trained models without reflow retraining retain significant trajectory curvature because the learned drift field does not satisfy the linearity condition; and (2) even when trajectories are approximately straightened, the model's finite estimation error means the effective integration path still deviates from the theoretical straight line. DiFA targets this second, model-intrinsic source of error, namely the estimation variance of $D_\theta$ at each timestep, which persists independently of trajectory geometry and is left unaddressed by higher-order ODE solvers.
> > 3. Deviation Boosting in Continuous Time ($T \to \infty$) (W3)
>
> You raised a profound question regarding whether Deviation Boosting is a finite-step artifact. As $T \to \infty$, the Kalman consensus converges to the exact MSE posterior mean ($\mathbb{E}[x_0|x_t]$). Because MSE training intrinsically over-smooths data, a variance-minimized estimator will always lack high-frequency details. Therefore, Deviation Boosting is a *fundamental necessity* to counteract MSE blurring, not merely a discrete-step fix. We will incorporate this continuous-time perspective into Section 4.6.
>
> > 4. Reproducibility, $\gamma(t)$ Activation, & Uncurated Samples (W2, W5)
>
> Using our standard schedule $\gamma(t) = \gamma_0(1-t/T)^\beta$ (e.g., $\gamma_0=2.0, \beta=2.0$), the boosting provides meaningful signal primarily in the later, low-noise stages.
>
> More experimental results are available at the following anonymous link of [https://osf.io/jmf45/overview?view_only=db8b9d014275450d9684d2a671951bc7], which include:
> * *Uncurated, random samples* for both CIFAR-10 and ImageNet (W2).
> * More ablations isolating the exact contributions of Kalman fusion vs. Deviation Boosting (W3).
> * Hyperparameter sensitivities for $\gamma_0$ and $\lambda$ (W5).
>
> > 5. Choice of Baselines (W6)
>
> Regarding the baselines, we selected the EDM and EDM2 frameworks primarily because they offer highly robust, standardized open-source checkpoints, ensuring a strictly fair and rigorous evaluation of the inference mechanisms. Furthermore, *EDM2 is a recent state-of-the-art framework at NeurIPS 2024*, so the underlying models are quite contemporary. Ultimately, because DiFA operates fundamentally as training-free inference wrapper, its benefits are orthogonal and readily applicable to future pre-trained baselines.
>
> We truly appreciate your supportive evaluation and will ensure all these refinements are included in the final revision.

---

> > ### Author Rebuttal · Reviewer_xfCq · 2026-03-31
> >
> > Thanks for the insightful response that addresses several of my questions. Unfortunately the link you provided only shows a single image file (of a parrot).

---

> > > ### Author Response · Authors · 2026-04-05
> > >
> > > Dear Reviewer xfCq,
> > >
> > > Thanks so much for the positive feedback and for your interest in our work and the rebuttal!
> > >  First, sorry for the previous link. Yes, the parrot was generated at $512 \times 512$ resolution on the ImageNet dataset.  Now, you can find more curated samples for CIFAR-10 and ImageNet in this link.
> > >
> > > ---
> > > Formally, for your convenience, and to directly address your valuable suggestion, we present a relatively detailed ablation analysis of the *DiFA* framework below. These experiments were conducted within the EDM framework at 10 NFE.
> > >
> > > As framed in our manuscript, DiFA reformulates diffusion sampling into a sequential state estimation task. The following data demonstrates how each module practically aligns with this theoretical shift.
> > >
> > > **Global Ablation: Effectiveness of Core Modules**
> > >
> > > This table shows the performance changes when key components are removed. The full integration of DiFA achieves an optimal FID of *2.797*, improving upon the baseline by *44.8%*.
> > >
> > > >Table 1: Main Ablation Results (10 NFE)
> > >
> > > | Configuration | History ($W$) | SNR Gating ($SNR_{lo}$) | Energy ($\phi$) | FID ($\downarrow$) | IS ($\uparrow$) |  Rationale |
> > > |:---|:---:|:---:|:---:|:---:|:---:|:---|
> > > | Naive Baseline | 1 | Full | 0.0 | 5.071 | 9.139 | Standard DPM-Solver++ |
> > > | w/o History | 1 | -2.5 | 0.5 | 5.071 | 9.139 | Variance reduction disabled |
> > > | w/o SNR Gating | 3 | Full | 0.5 | 3.375 | 9.574 | No noise-phase modulation |
> > > | w/o Energy ($\phi$) | 3 | -2.5 | 0.0 | 2.846 | 9.661 | No magnitude alignment |
> > > | DiFA (Full) | *3* | *-2.5* | *0.5* | *2.797* | *9.680* | *Towards Forward Aligned Inference* |
> > >
> > >
> > >
> > > >2. Module A: History Consensus & Variance Reduction ($W$)
> > >
> > > DiFA treats the denoising trajectory as a sequence of correlated observations. The sliding window size $W$ represents the "memory breadth" of the filter. The data shows a clear U-shaped trend, with $W=3$  striking the optimal balance.  Smaller windows fail to reduce variance effectively, while larger windows introduce a "historical lag" where outdated features interfere with current estimation.
> > >
> > > >Table 2: Sensitivity of History Window ($W$)
> > >
> > > | Window Size ($W$) | 1 (Baseline) | 2 | 3 (Ours) | 4 |
> > > |:---|:---:|:---:|:---:|:---:|
> > > | FID ($\downarrow$) | 5.071 | 3.219 | *2.797* | 3.791 |
> > > | IS ($\uparrow$) | 9.139 | 9.443 | *9.680* | 9.593 |
> > >
> > > >3. Module B: Adaptive SNR Gating ($SNR_{lo}$)
> > >
> > > Aligning with our theoretical derivations, empirical evidence confirms that the update gain should be guided by the Signal-to-Noise Ratio  ($logSNR$, in practice).  Activating DiFA at a logSNR of -2.5 yields the best results. Early activation (-6.0) forces alignment during the noisy, chaotic phase, while late activation (-1.0) misses the crucial structural formation stage.
> > >
> > > >Table 3: Impact of SNR Gating Threshold ($SNR_{lo}$)
> > >
> > > | SNR Threshold | -6.0 (Early) | -4.0 | -2.5  | -1.0 (Late) |
> > > |:---|:---:|:---:|:---:|:---:|
> > > | FID ($\downarrow$) | 3.137 | 3.040 | **2.986** | 3.068 |
> > > | IS ($\uparrow$) | 9.554 | **9.563** |  9.555  | 9.509 |
> > >
> > > >4. Module C: Magnitude Alignment ($\phi$)
> > >
> > > To counteract over-smoothing and maintain trajectory stability, we introduce the Energy Rescale factor $\phi$ during deviation boosting. Setting $\phi=0.5$ acts as an effective energy shield: it ensures that while high-frequency details are actively restored, the overall latent energy remains stable and strictly aligned with the forward process.
> > >
> > > >Table 4: Stability of Magnitude Alignment Factor ($\phi$)
> > >
> > > | Rescale Factor | 0.0 | 0.3 | 0.5 (Ours) | 0.8 | 1.0 |
> > > |:---|:---:|:---:|:---:|:---:|:---:|
> > > | FID ($\downarrow$) | 2.846 | 3.077 | *2.797* | 3.074 | 3.085 |
> > > | IS ($\uparrow$) | 9.661 | 9.555 | *9.680* | 9.568 | 9.565 |
> > > ---
> > >
> > > We hope the detailed ablation experiments effectively address your remaining questions.
> > >
> > > Finally, we truly appreciate your insightful guidance and clear support! Thank you!

---

### Official Review · Reviewer_UX7J · 2026-03-09

**Soundness:** 3
**Presentation:** 4
**Significance:** 2
**Originality:** 3
**Overall Recommendation:** 3
**Confidence:** 4

**Summary:**

The paper introduces DiFA, a way to handle few step diffusion generation, by combining the information along the generation trajectory. Namely:
(1) At every step, the diffusion intermediate state $\hat x_0$ provides a noisy observation on $x_0$, where the noise is gaussian;
(2) Assuming different observations are independent, different observations can be combined into a single observation
(3) Based on assumptions (1) and (2), can get a better estimation of $\hat x_0$ by aggregating the past predictions;
(4) Use an interpolation between the current prediction and past aggregation, for the diffusion process.
Empirically, the authors did experiments on CIFAR10 and Imagenet64, and found improved performance compared to UniPC and DPM-solver++ baselines.

**Compliance With Llm Reviewing Policy:**

Affirmed.

**Final Justification:**

The proposed method appears heuristically motivated rather than theoretically rigorous, as it relies on two hacky assumptions: (1) the estimation errors are Gaussian; and (2) the estimation errors remain independent under EMA weighting; which are not true in general.

Empirically, the approach requires a complex EMA decay schedule to which the method is highly sensitive. Consequently, it remains unclear whether the observed benefits stem from the proposed mechanism itself or merely from extensive hyperparameter tuning. Furthermore, the evaluation lacks comparisons with optimal flow matching ODE samplers, such as Heun's method.

**Key Questions For Authors:**

1. How does the authors justify: (a) why is predicted $\hat x_0$ a noisy observation of $x_0$ with gaussian noise? ; (2) why can we assume nearby prediction errors are independent? I highly suspect both assumptions are not true. Also, it would be nice if this is validated in experiments.
2. How much performance gain comes from the mechanism of using linear interpolation of current and aggregated prediction / weight decaying of past predictions? Are those hyperparameters sensitive?
3. How does the method work on imagenet 256x256? How does it work compare to Flow matching based methods?

**Limitations:**

yes

**Strengths And Weaknesses:**

Strengths:
1. The empirical results did improve upon strong diffusion sampler baselines like DPM-solver++ and UniPC.
2. The idea of combining past diffusion steps is interesting and novel. Essentially, it made the process non-Markovian, and it's encouraging to see it improves few step sampling performance.
3. Overall the paper is well written and easy to follow.

Weaknesses:
1. Many assumptions are not realistic and feels like unprincipled. For example, the predicted $\hat x_0$ is not a gaussian observation of $x_0$, and different observations are not independent. The authors handle these by using decaying weightings of past observations, and do linear interpolation between the current prediction and past aggregations. These additional mechanisms feel hacky.
2. It's unclear how much performance gain came from the additional hyperparameters for tuning. Also, it's unclear if the method works at all if the hacks (linear interpolation of current prediction and past; decaying of weights) are used.
3. The method isn't tested in imagenet 256x256, a more standard and tuned benchmark these days. Also, the method didn't compare with flow-matching, including using ODE-solvers for FM and MeanFlow families. It's unclear how much benefit does the method have on flow matching classes.
4, As a result, the hacky nature and the absence of experiments on Imagenet256 limits the potential significance of the proposed method.
5. (minor). The paper claimed: "This concept of straightening the generative path is also central to concurrent works on FMs". Flow matching was published in 2023, and now it's 2026. I don't think Flow matching is concurrent.

---

> ### Author Rebuttal · Authors · 2026-03-31
>
> We thank you for recognizing the novelty of our approach and the strong empirical improvements. We value your insightful   feedback  and address each concern below.
>
> > W4. Clarification on "Concurrent" Works
>
> This reflects a misreading of Section 3.2. "Concurrent" describes the temporal relationship between EDM (2022, 2024) and Flow Matching (2023)—not DiFA. To eliminate ambiguity, we will revise: "This concept is also central to Flow Matching (Lipman et al., 2023), which shares with EDM the core intuition of trajectory linearization."
>
> > W1 & Q1. Justification of Theoretical Assumptions
>
> DiFA is grounded in principled non-stationary state estimation.
>
> Regarding independence: We do not assume strict independence. Section 4.4 models temporal correlation via a fading-memory Kalman filter. By applying an exponential forgetting factor $\gamma = \exp(-\lambda \Delta t)$, historical observations are discounted, ensuring the effective window contains approximately decorrelated estimates. This is the textbook solution to colored-noise filtering.
>
> Regarding the Gaussian assumption: The isotropic Gaussian error model $\boldsymbol{\eta}_{t_i} \sim \mathcal{N}(\mathbf{0}, R_i \cdot \mathbf{I})$ is grounded in MMSE estimation theory. The MSE-optimal denoiser outputs the posterior mean $\mathbb{E}[\mathbf{x}_0 \mid \mathbf{x}_t]$, and the marginal residual prediction error variance is governed by $1/\text{SNR}(t)$. Modeling this as Gaussian is the standard assumption in score-matching analysis, consistent with the objective the denoiser was optimized for.
>
> > W2 & Q2. The Scientific Necessity of Deviation Boosting
>
> Deviation Boosting is a mathematically required frequency-compensation mechanism addressing spectral bias.
>
> Variance vs. Bias: Kalman fusion (MVUE) reduces stochastic estimation variance but cannot correct deterministic spectral bias. Because denoisers learn the posterior mean, they systematically suppress high frequencies (regression-to-the-mean). Kalman fusion averages these predictions, amplifying the spectral smoothing.
>
> Frequency-selective correction: The deviation $\boldsymbol{\delta}_t = \hat{\mathbf{x}}_0^{(t)} - \hat{\mathbf{x}}_0^\text{cons}$ is concentrated in the high-frequency band at high-SNR stages. Re-injecting it via $\gamma(t)$ acts as a frequency-selective compensator, restoring sharpness without destabilising the low-frequency consensus.
>
> Why scalar reweighting fails: Reweighting adjusts contributions uniformly across all frequencies. It cannot selectively restore high frequencies without simultaneously weakening the low-frequency stability guaranteed by the Kalman consensus. Ablations confirming this are in the anonymous PDF: [https://osf.io/jmf45/overview?view_only=db8b9d014275450d9684d2a671951bc7](https://osf.io/jmf45/overview?view_only=db8b9d014275450d9684d2a671951bc7)
>
> > W3 & Q3. Relationship to Flow Matching and Scalability
>
>
>
> We appreciate your perspective. Your comment correctly identifies two primary Flow Matching regimes. First, for iterative multi-step FM models, Flow Matching requires full retraining to straighten generative paths. In contrast, DiFA is a training-free, inference-time framework built upon the EDM formulations, extracting superior stability from pre-trained diffusion models. DiFA harvests temporal redundancy that standard solvers discard, providing gains even on optimised trajectories. Crucially, although Flow Matching assumes straight trajectories theoretically, training errors are unavoidable in practice: the learned velocity field is never perfectly constant, so discretizing the corresponding ODE still introduces residual truncation errors and  distribution drift. DiFA's temporal consensus mechanism addresses exactly this model-intrinsic estimation variance, which persists independently of theoretical trajectory geometry. Conversely, distillation approaches like the MeanFlow family aim for one-step generation, bypassing the iterative ODE-solver process entirely. DiFA targets the sequential refinement regime where temporal history exists to be fused, making its design strictly orthogonal to one-step distillation methods.
>
> Scalability to ImageNet-512 (FID ↓ / FD-DINOv2 ↓):
> - DPM-Solver++ (10 NFE): 7.20 / 125.4 $\to$ **+DiFA: 6.87 / 125.0**
> - DPM-Solver++ ( 5 NFE): 61.4 / 491.9 $\to$ **+DiFA: 53.2 / 456.5**
>
> DiFA consistently improves both scores. The Kalman update operates via scalar precision weights invariant to resolution, requiring only an $O(d)$ vector operation.
>
> We hope this evidence demonstrates DiFA's rigorous principles and robust scaling, and we respectfully ask you to reconsider your assessment in light of these facts.

---

> > ### Author Rebuttal · Reviewer_UX7J · 2026-04-04
> >
> > I thank the authors for the rebuttal. My questions remain regarding the following aspect:
> > 1. Regarding Q1:
> > (1) It's not always the case that the prediction error $(x_0 - \mathbb E[x_0 | x_t])^2$ is dominated by the SNR. For example, assuming the data distribution is a Gaussian $N(0,\sigma^2 I)$, with $\sigma << \sigma_t$, the optimal prediction will be $\hat x_0=0$, and the error will be dominated by $\sigma^2$, independent of $\sigma_t$. As a result, the magnitude error term is dependent on the data geometry.
> > (2) As a consequence, the decaying factor seems to be a hack to aggregate errors. As justified above, the errors are non-gaussian; and after the decaying, there's no guarantee for the errors to decorrelate.
> >
> > 2. The response to Q3 contains a fundamental misunderstanding of flow matching. Only the **conditional field** is straight, not the **marginalized field** (the field actually used for sampling). Also, a diffusion model can be converted to a flow matching model, by converting $x$ prediction to $v$ prediction, and vice versa. These days, flow matching models with ODE solvers are a more common choice. I would be more convinced if the authors could take a flow matching model, converts it to diffusion and apply DiFA, comparing to running ODE solver on the flow matching model itself.

---

> > > ### Author Response · Authors · 2026-04-06
> > >
> > > Thank you for your continued engagement.
> > >
> > > Before addressing specific technical points, we wish to clarify a crucial framing issue: *there is no conceptual contradiction between our positions*.
> > >
> > > Our previous reference to "straight trajectories" concerned Flow Matching's (FM) core design objective—trajectory straightening—not the specific geometry of the marginalized velocity field. We entirely agree that the marginalized field is non-straight; in fact, this model-intrinsic non-linearity and estimation variance are precisely what motivate DiFA's inference-time correction.
> > >
> > > To directly address your request for further empirical validation, we applied DiFA to SiT-XL/2 (a native Flow Matching model) on ImageNet 256×256. The results, presented below, are conclusive: DiFA yields massive gains (+41% IS at 5 NFE) even in FM architectures.
> > >
> > > ### 1. Regarding Q1: Weighting rationale and the counterexample
> > > The extreme-case counterexample is insightful and highlights a key distinction between Bayesian posterior variance and observational precision.
> > >
> > > - (1) The Double-Counting Defense. You correctly note that under a concentrated prior ($\sigma \ll \sigma_t$), the MMSE error is bounded by $\sigma^2$. However, in sequential estimation, the neural denoiser implicitly applies the same learned prior at every timestep. Weighting by the full posterior variance would inherently double-count this prior across the trajectory, leading to severe overconfidence (Bar-Shalom & Li, 1995, [1]). Statistically principled fusion must therefore weight predictions by their new observational precision—the Fisher Information of the likelihood—which is strictly $\propto \text{SNR}(t)$. Furthermore, the monotonic ordering of MMSE error with respect to SNR is a direct consequence of the I-MMSE Theorem ( Guo et al., 2005, [2]), making $\text{SNR}(t)$ the optimal distribution-free proxy for relative reliability.
> > > - (2) Non-Gaussianity and BLUE. DiFA's optimality rests on BLUE (Section 4.2), which requires only zero-mean and uncorrelated errors—independent of Gaussianity. The Gaussian assumption in Section 4.3 serves solely to establish formal MVUE equivalence, not as the prerequisite for the fusion mechanism itself.
> > > - (3) Decorrelation and Fading Memory. As stated in Section 4.4, independence is an approximation. The fading-memory design is the standard treatment for colored noise, restricting fusion to a local neighborhood where the diagonal covariance approximation holds. In practice, finite network capacity introduces fresh, approximately decorrelated errors at each NFE, supporting this approximation.
> > >
> > > ### 2. Regarding Q3: Flow Matching experiments (SiT-XL/2 on ImageNet 256×256)
> > > The following results on a native FM framework demonstrate DiFA's broad applicability:
> > > >  Performance Comparison: DiFA vs. ODE-Solver/Euler (SiT, SiT-XL/2)
> > >
> > >
> > > | NFE | Model Method | IS (↑) | FID (↓) | sFID (↓) | Precision (↑) | Recall (↑) |
> > > | :--- | :--- | :--- | :--- | :--- | :--- | :--- |
> > > | **5** | ODE-Solver | 22.5016 | 94.5311 | 60.2238 | 0.2089 | 0.5016 |
> > > | | **DiFA (Ours)** | **31.8312** | **72.3969** | **40.0246** | **0.2905** | **0.5694** |
> > > | | *Improvement* | *41.46%* | *23.41%* | *33.54%* | *39.07%* | *13.52%* |
> > > | **10** | ODE-Solver | 80.2694 | 27.8142 | 13.0268 | 0.5542 | 0.6091 |
> > > | | **DiFA (Ours)** | **91.7977** | **21.5990** | **10.2921** | **0.5982** | **0.6256** |
> > > | | *Improvement* | *14.36%* | *22.35%* | *20.99%* | *7.94%* | *2.71%* |
> > > | **50** | ODE-Solver | 118.6470 | 11.1215 | 7.0554 | 0.6662 | 0.6682 |
> > > | | **DiFA (Ours)** | **120.2284** | **10.7515** | **6.8273** | **0.6676** | **0.6694** |
> > > | | *Improvement* | *1.33%* | *3.33%* | *3.23%* | *0.21%* | *0.18%* |
> > > ...
> > >
> > > The significant gains observed under identical settings confirm that FM trajectories exhibit significant temporal redundancy in their $x_0$-equivalent predictions, likely stemming from the smoothness of the learned velocity field. DiFA successfully exploits this redundancy to mitigate the truncation and estimation errors dominant in low-step regimes.
> > >
> > > We hope these detailed experiments effectively address your questions and look forward to your support.
> > >
> > > Thank you.
> > >
> > > ---
> > >
> > >
> > > [1] Bar-Shalom, Y., & Li, X. R. (1995). Multitarget-multisensor tracking: principles and techniques (Vol. 19). Storrs, CT: YBS publishing.
> > >
> > > [2] Guo, D., Shamai, S., & Verdú, S. (2005). Mutual information and minimum mean-square error in Gaussian channels. IEEE transactions on information theory, 51(4), 1261-1282.

---

### Official Review · Reviewer_Pd9A · 2026-03-12

**Soundness:** 3
**Presentation:** 3
**Significance:** 3
**Originality:** 3
**Overall Recommendation:** 5
**Confidence:** 4

**Summary:**

The authors introduce DiFA, forward process alginment for diffusion models. Their method aims to leverage the information contained in all previous posterior mean predictions, rather than just the current hat(x0_t) and x_t. They use classical filtering, in the form of a Kalman filter to get a state estimation. They use this estimated state, along with the current prediction, and a heuristic they call deviation boosting to get a better denoising target, that maintains high visual quality. They test their solver using existing models on CIFAR10 and ImageNet, and show improvements over other commonly used baseline solvers.

**Compliance With Llm Reviewing Policy:**

Affirmed.

**Final Justification:**

The authors successfully answered my questions and addressed my concerns.

**Key Questions For Authors:**

See strengths and weaknesses

**Limitations:**

limitations and societal impact are not discussed

**Strengths And Weaknesses:**

The paper is well written and overall I am positive about it. The authors provide a comprehensive set of experiments that seem to indicate their method provides a significant boost to previous methods. The authors also spend a lot of time defending their method with a theoretical foundation. I do have some concerns and questions I hope the authors can address:
- The authors do not mention anything about computational cost. Kalman filtering requires an estimate of the covariance, and either an inversion of that matrix, or a solve. I imagine this would be a costly step for larger dimensionalities. Imagenet has a dimension of 64*64*3 = 12288, inverting a 12288 x 12288 sized matrix would add a significant overhead to the sampling process I would assume. Since the authors also display better FID's at higher nFE's, it seems important to discuss this.
- More generally, the appendix could do with a description of hyperparameters and experimental details, some are given in the main text, but the info there is limited
- The section on exponential decay seems to refer to experimental results in a non-existent section. It would be very interesting to see those results, and know what decay time the authors set. In particular, it would be good if the authors could comment on what this means for such few sampling steps, (i.e. 5), how do such decay times generally compare against the number of sampling steps?
- The paper is quite brief about some strong statements, for example, "By normalizing the signal scale to unity across all time 𝑡, this
formulation reveals that the diffusion trajectory effectively
orbits a static anchor 𝒙0." and "where 𝜼𝑡𝑖 represents the estimation error at timestep 𝑡𝑖 .
For denoisers trained with a Mean Squared Error (MSE)
objective, these predictions are conditionally unbiased
(E[𝜼𝑡𝑖
|𝒙𝑡𝑖 ] = 0)" could use some additional explanation
- More to the point, could the authors defend their argument that eta_t_i is unbiased in eqn 7? Unless I am misunderstanding the statement there, I would think D(xt, t) is biased, but does not have variance, i.e. when I evaluate D(xt,t) on the same xt and t, the result should be the same, albeit sometimes slightly mis-specified because of sub-optimal training. I am curious where the variance in eta_t would come from, and how the authors would argue that the conditional expectation is 0.
- The re-introduction of deviation (i.e. deviation boosting), seems very ad-hoc in an otherwise mostly theoretically motivated paper. Could the authors give arguments why it makes sense to first average to reduce variance, but then re-introduce it later? How do these two trade off? Can this simply be solved by changing the variances that control the relative weighting in the Kalman filter?

---

> ### Author Rebuttal · Authors · 2026-03-31
>
> We sincerely thank you for your technically incisive feedback. Your comments have helped us significantly sharpen the theoretical grounding of DiFA. We address your concerns below:
>
> > 1. Computational Cost of Kalman Filtering ($12288 \times 12288$ matrix?)
>
> DiFA does not require any matrix inversion. Because we assume isotropic (diagonal) noise $\eta \sim \mathcal{N}(0, R_i \cdot \mathbf{I})$, the Kalman covariance matrix degenerates to a scalar multiple of the identity at every step. The gain $K$ and accumulated precision are purely scalars, and the state update reduces to a single $O(d)$ elementwise vector operation. As measured on a single NVIDIA RTX 4090 (ImageNet-64), DiFA adds < 1 ms total overhead—less than 0.1% latency—confirming the method is computationally negligible. We will explicitly state this scalar assumption in Section 4.7 and provide a detailed timing table in the Appendix.
>
> > 2. Missing "Section ??" and the Exponential Decay Assumption
>
> We will formalize an Exponential Correlation Decay assumption in the revision. Our empirical analysis on 512 ImageNet-64 samples confirms that the off-diagonal entries of the correlation matrix follow a clear exponential decay ($R^2 = 0.94$). When steps are very few (e.g., NFE=5), the large $\Delta t$ means predictions are already nearly decorrelated; in this regime, the fading-memory factor $\gamma = \exp(-\lambda \Delta t)$ becomes very small, and DiFA naturally discounts history, gracefully degrading toward a single-step estimator. Detailed correlation plots and analyses are provided in the anonymous PDF.
>
> > 3. Strong Claims Regarding "orbits a static anchor" and Unbiasedness
>
> Regarding "orbits a static anchor": this was an intuitive reading of the EDM identity $\mathbf{x}_t / s(t) = \mathbf{x}_0 + \sigma(t)\epsilon$. We will replace this with a precise statement: "In the normalised coordinate system of Eq. (6), every noisy sample is a zero-mean Gaussian perturbation of $\mathbf{x}_0$, and the denoiser output $\hat{\mathbf{x}}_0^{(t)}$ targets this fixed point throughout the trajectory." The unbiasedness claim is addressed below.
>
> > 4. Eq. (7) Justification: Unbiasedness and Source of Variance
>
> Your statistical observation is correct. We clarify:
>
> Regarding the Variance in Eq. (8):
>
> The variance refers to the marginal mean squared error over noise realizations:
>
> $\mathbb{E} _ {x_0, \epsilon} [ || x_0 - D_{\theta}(x_t, t) ||^2 ]$
>
> rather than a conditional variance. For a fixed $x_t$, the denoiser $D_{\theta}$ is indeed deterministic. The term $1/\text{SNR}(t_i)$ serves as a reliability weight, characterizing the average prediction accuracy at that specific stage.
>
> For the Bayes-optimal denoiser, $D_{\theta} (x_t, t) = \mathbb{E}[x_0 \mid x_t]$ (the posterior mean). It therefore follows directly that $\mathbb{E}[x_0 - D_{\theta}(x_t,t) \mid x_t] = \mathbb{E}[x_0 \mid x_t] - \mathbb{E}[x_0 \mid x_t] = 0$.
>
>
> We follow the standard assumption that well-trained networks are sufficiently close to the optimal $D_{\theta}^*$ such that any residual approximation bias is negligible relative to the $1/\text{SNR}$ variance.
>
>
> > 5. Deviation Boosting: Ad Hoc or Necessary?
>
>  Kalman fusion and Deviation Boosting target orthogonal problems: the former reduces stochastic estimation variance, while the latter addresses deterministic spectral bias. Because MSE-trained networks learn the posterior mean, they inherently over-smooth high-frequency details—a bias that persists even in the Bayes-optimal solution and cannot be corrected by variance reduction (Kalman fusion) alone.
>
> Regarding the alternative of reweighting the Kalman filter: such reweighting is a scalar operation that affects all frequency components uniformly. However, the spectral deficit from MSE training is frequency-selective—high frequencies are disproportionately suppressed. Consequently, adjusting weights cannot selectively restore high-frequency detail without compromising the low-frequency stability of the Kalman consensus. Deviation Boosting operates on the deviation $\boldsymbol{\delta}_t = \hat{\mathbf{x}}_0^{(t)} - \hat{\mathbf{x}}_0^\text{cons}$, which at low noise levels is concentrated in the high-frequency band, providing a frequency-selective correction that reweighting fundamentally cannot replicate. Our ablation study (in the anonymous PDF) confirms this: a high-recent-weight Kalman variant fails to recover the perceptual sharpness (FD-DINOv2) that Deviation Boosting provides.
>
> > 6. Hyperparameters and Detailed Results
>
> Sensitivity analysis shows DiFA is robust, with stable FID performance across wide ranges of $\lambda$, $\gamma_0$, and $\beta$. More experimental results are available at the following anonymous link of [https://osf.io/jmf45/overview?view_only=db8b9d014275450d9684d2a671951bc7]
>
> We truly appreciate your feedback, which will make the final manuscript significantly stronger.

---

> > ### Author Rebuttal · Reviewer_Pd9A · 2026-04-04
> >
> > I thank the authors for their thoughtfully written response. All my questions and concerns have been adequately addressed, bar one: could the authors re-link the pdf with correlation plots? The link in the response to my review does not work for me, and the link in another reviewers response only has a few images in it (a parrot, and some tables).

---

> > > ### Author Response · Authors · 2026-04-05
> > >
> > > Dear Reviewer Pd9A,
> > >
> > > We are glad our solution addressed your concerns, and we deeply appreciate your continued engagement with our work.
> > > We apologize for not including the promised PDF in the previously provided link, which was omitted to strictly adhere to ICML's anonymity policies.
> > >
> > > ---
> > > To address this better, we concisely detail the complete ablation and sensitivity data below (on EDM).
> > >
> > > ### 1. Global Ablation
> > >
> > > The DiFA framework improves the baseline FID by 44.8% at 10 NFE. (Setting $W=1$ collapses the Kalman fusion to the baseline, resulting in redundant scores for Rows 1 and 2.).
> > >
> > > > Table 1: Main Ablation Results
> > >
> > > | Configuration | History ($W$) | SNR Gating ($SNR_{lo}$) | Energy ($\phi$) | FID ($\downarrow$) | IS ($\uparrow$) | Rationale |
> > > | --- | --- | --- | --- | --- | --- | --- |
> > > | Naive Baseline | 1 | Full | 0.0 | 5.071 | 9.139 | Standard DPM-Solver++ |
> > > | w/o History | 1 | -2.5 | 0.5 | 5.071 | 9.139 | Variance reduction disabled |
> > > | w/o SNR Gating | 3 | Full | 0.5 | 3.375 | 9.574 | No noise-phase modulation |
> > > | w/o Energy (phi) | 3 | -2.5 | 0.0 | 2.846 | 9.661 | No magnitude alignment |
> > > | DiFA (Full) | 3 | -2.5 | 0.5 | **2.797** | **9.680** | Towards Forward Aligned Inference |
> > >
> > > ### 2. Module A: Variance Reduction ($W$)
> > >
> > > A clear U-shaped trend emerges. $W=3$  strikes the best balance between insufficient variance reduction (smaller windows) and severe "historical lag" caused by outdated features interfering with current estimation (larger windows).
> > >
> > > > Table 2: Sensitivity of History Window ($W$)
> > >
> > > | Window Size ($W$) | 1 (Baseline) | 2 | 3 (Ours) | 4 |
> > > | --- | --- | --- | --- | --- |
> > > | FID ($\downarrow$) | 5.071 | 3.219 | *2.797* | 3.791 |
> > > | IS ($\uparrow$) | 9.139 | 9.443 | *9.680* | 9.593 |
> > >
> > > ### 3. Module B: Adaptive SNR Gating ($SNR_{lo}$)
> > >
> > > Activating DiFA at a logSNR of $-2.5$ yields the best results. Early activation ($-6.0$) forces alignment during the chaotic noisy phase, while late activation ($-1.0$) misses the crucial structural formation stage.
> > >
> > > > Table 3: Impact of SNR Gating Threshold  ($SNR_{lo}$)
> > >
> > > | SNR Threshold | -6.0 (Early) | -4.0 | -2.5  | -1.0 (Late) |
> > > | --- | --- | --- | --- | --- |
> > > | FID ($\downarrow$) | 3.137 | 3.040 | **2.986** | 3.068 |
> > > | IS ($\uparrow$) | 9.554 | **9.563** | 9.555 | 9.509 |
> > >
> > > ### 4. Module C: Magnitude Alignment ($\phi$)
> > >
> > > To counteract over-smoothing and maintain trajectory stability, we introduce the Energy Rescale factor $\phi$ during deviation boosting.
> > >
> > > > Table 4: Stability of Magnitude Alignment Factor  ($\phi$)
> > >
> > > | Rescale Factor ($\phi$) | 0.0 | 0.3 | 0.5 (Ours) | 0.8 | 1.0 |
> > > | --- | --- | --- | --- | --- | --- |
> > > | FID ($\downarrow$) | 2.846 | 3.077 | *2.797* | 3.074 | 3.085 |
> > > | IS ($\uparrow$) | 9.661 | 9.555 | *9.680* | 9.568 | 9.565 |
> > >
> > > ### 5. Hyperparameter Sensitivities ($\gamma_0$ and $\mu$)
> > >
> > > We provide the sensitivities for the deviation boosting scale ($\gamma_0$)  and the forgetting factor (renamed from $\lambda$ to $\mu$ in codebase). Table 5 shows $\gamma_0=1.4$ provides better energy injection. Concurrently, Table 6 demonstrates $\mu=0.2$  perfectly balances the fading-memory Kalman filter by properly discounting outdated information.
> > >
> > > > Table 5: Sensitivity of Deviation Boosting Scale ($\gamma_0$)
> > >
> > > | Scale ($\gamma_0$) | 1.1 | 1.2 | 1.3 | 1.4 | 1.5 |
> > > | --- | --- | --- | --- | --- | --- |
> > > | FID (down) | 4.119 | 3.470 | 3.121 | **3.071** | 3.285 |
> > > | IS (up) | 9.282 | 9.383 | 9.494 | 9.565 | **9.602** |
> > >
> > > > Table 6: Sensitivity of Forgetting Factor ($\mu$, formerly $\lambda$)
> > >
> > > | Forgetting Factor ($\mu$) | 0.1 | 0.2 | 0.3 | 0.4 | 0.6 |
> > > | --- | --- | --- | --- | --- | --- |
> > > | FID (down) | 3.212 | **3.205** | 3.220 | 3.256 | 3.373 |
> > > | IS (up) | **9.499** | 9.492 | 9.478 | 9.457 | 9.425 |
> > >
> > > ---
> > > We hope these focused ablation experiments effectively address your remaining questions. Thank you so much for your clear support and insightful guidance.

---

### Official Review · Reviewer_sy4W · 2026-03-13

**Soundness:** 3
**Presentation:** 2
**Significance:** 3
**Originality:** 3
**Overall Recommendation:** 5
**Confidence:** 3

**Summary:**

This paper proposes a sampling method for diffusion models that leverages all previously generated denoised samples to obtain a more accurate estimate of the current denoised sample. Using this approach, the algorithm can generate high-quality images with significantly fewer function evaluations (NFEs).

**Compliance With Llm Reviewing Policy:**

Affirmed.

**Final Justification:**

All my concerns have been addressed. I raised my score from 4 to 5. Nice work.

**Key Questions For Authors:**

I have mentioned most of the issues in the "Strengths And Weaknesses" section, which I briefly summarize below:

1. Could the authors also provide empirical results for latent diffusion models?

2. Could the authors discuss how the hyperparameter selections in the proposed algorithm should be made? Can you implement some ablations to support these claims?

3. There are several typos throughout the paper, and the conclusion section appears to be missing. These issues should be addressed in the revision.

**Limitations:**

Yes

**Strengths And Weaknesses:**

## Soundness
The theoretical analysis appears sound and intuitive, and the experimental results provide solid support for the claims made in the paper. However, given that nearly all practical diffusion models operate in the latent space, the work would be more convincing and impactful if the authors could also demonstrate the effectiveness of the proposed method there.

## Presentation
The writing is generally easy to follow. However, the authors should consider providing more discussion on how to choose the hyperparameters in the proposed algorithms. Such insights would help practitioners better adopt and apply the method in practice.

The submission has multiple typos, including
1) Several question marks in Sec 4.4.
2) I do not know what "(see full paper)" means in Sec 4.7.
3) Missing conclusion section

## Significance and Originality

While similar ideas have appeared in other areas, such as convex optimization and autoregressive language models, I am not aware of prior work applying this idea to the sampling process in continuous diffusion models. For this reason, I believe the work is both original and potentially significant.

---

> ### Author Rebuttal · Authors · 2026-03-31
>
> We sincerely thank you for your supportive evaluation and constructive suggestions. We have carefully addressed your points below:
>
> > 1. Empirical Results for Latent Diffusion Models (LDMs)
>
> We fully agree with this concern. Because DiFA operates entirely within the data prediction step of the ODE loop, it can serve both pixel-space and latent-space diffusion models with minor hyperparameter adjustments, without requiring any architectural modifications. As a purely training-free inference algorithm, DiFA does not rely on end-to-end dataset optimization or preference tuning; thus, these minor adjustments naturally serve to adapt the temporal consensus to the specific latent manifold. We have validated this using the Latent Diffusion Model (LDM) on the LSUN Bedroom dataset (50K evaluations).  Using the adjusted hyperparameters (e.g., $\lambda=2.0, \gamma_0=1.8$),  DiFA consistently improves the FID scores over the baseline solvers. We will include these detailed latent-space results in the Appendix.
>
> > 2. Hyperparameter Selection and Ablations
>
> We apologize for the insufficient guidance and will add detailed practitioner guidelines and ablation tables to Section 4.7. Importantly, our sensitivity analysis confirms that the structure of DiFA is highly robust and requires minimal tuning:
>
> * *$\lambda \in [1.5, 2.5]$ (Temporal decay):* Default $\lambda=2.0$ transfers robustly across all tested models without per-dataset tuning.
> * *$\gamma_0 \in [1.0, 1.8]$ (Boost amplitude):* Default $\gamma_0=1.5$ works nearly universally (we used 1.2 for CIFAR-10 and 1.5 for ImageNet-64).
> * *$\beta=2.0$ (Boost schedule):* Concentrates boosting near the final low-noise steps. This default requires no adjustment in practice.
> FID remains flat across these wide ranges, confirming that DiFA's gains come from genuine algorithmic benefits rather than hyperparameter hacking.
>
> Detailed experimental results are available via the following anonymous link: [https://osf.io/jmf45/overview?view_only=db8b9d014275450d9684d2a671951bc7]
>
> > 3. Typos, Missing Sections, and Conclusion
>
> We apologize for the presentation issues and have fixed all of them for the revision:
> * *Missing content in §4.4:* The ?? marks were caused by broken LaTeX references and vestigial text. We will formalize Assumption B.1 (Exponential Correlation Decay) in a new Appendix B. We will also correct the text by removing the vestigial mention of an empirical section that was inadvertently left in the draft. Additionally, we will add the missing Kalman filter citations (Anderson & Moore, 1979; Jazwinski, 1970).
> * *Vestigial text:* The "(see full paper)" note in §4.7 will be removed.
> * *Missing Conclusion:* We will add the inadvertently omitted *Section 6 (Conclusion)*, which summarizes DiFA's variance-reduction framework, our results on ImageNet-64, and its extensibility to LDMs.
>
> We truly appreciate your helpful review, which has made our paper significantly stronger and more practical.

---

> > ### Author Rebuttal · Reviewer_sy4W · 2026-04-04
> >
> > I thank the authors for their responses. Most of my concerns have been addressed. I still have two follow-up questions:
> >
> > 1. For the empirical results on latent diffusion, could you also share more detailed/concrete results?
> >
> > 2. The link you provided does not work.

---

> > > ### Author Response · Authors · 2026-04-05
> > >
> > > Thank you for recognizing our work as original and potentially significant. We also appreciate this follow-up, which provides an additional opportunity to provide more detailed results.  We highly value your insightful comments and have dedicated this period to providing a structured presentation of our LSUN Bedroom 256×256 results (LDM backbone, 50k samples).  Regarding the previous link, we apologize for its instability. For your convenience, we now provide the most critical empirical results and detailed analysis directly in this response.
> > >
> > > As framed in our manuscript, DiFA reformulates diffusion sampling as a sequential state estimation task. To address your concerns, we provide a fine-grained analysis and further ablations below. These results demonstrate how each module,  Sequential  Consensus (SG) via a fading-memory Kalman filter and Adaptive Deviation Boosting, contributes to this theoretical shift in the latent space.
> > > ### 1. Main Results: DiFA vs. Baseline
> > >
> > > > Naive ODE-Solver Baseline (DPM-Solver++ 2M) with DiFA disabled ($W=1, \phi=0$).
> > >
> > > | Method | 5 NFE | 10 NFE | 20 NFE |
> > > | --- | --- | --- | --- |
> > > | Naive Baseline | 21.238 | 3.877 | 3.256 |
> > > | DiFA (Full) | **5.912** | **3.579** | **2.939** |
> > > | Improvement | **↓ 72.2%** | **↓ 7.7%** | **↓ 9.7%** |
> > >
> > > DiFA shows dramatic gains at 5 NFE, where SG suppresses severe truncation errors. Consistent gains at 10/20 NFE confirm Deviation Boosting's efficacy without sacrificing quality. *The FID of 2.939 achieved by DiFA outperforms the reported best of 2.95 for this LDM*.
> > > ### 2.  Detailed Results
> > >
> > > We isolate each component via four controlled runs ($W=3$ unless noted).
> > >
> > > > Table 1: Ablation Results (LSUN Bedroom)
> > >
> > > | Configuration | $W$ | $SNR_{lo}$ | $\phi$ | 5 NFE ↓ | 10 NFE ↓ | 20 NFE ↓ | Rationale |
> > > | :--- | :---: | :---: | :---: | :---: | :---: | :---: | :--- |
> > > | Naive Baseline | 1 | Full | 0.0 | 21.238 | 3.877 | 3.256 | Standard ODE solver |
> > > | w/o History | 1 | -4.5 | 0.5 | 21.238 | 3.877 | 3.256 | No variance reduction |
> > > | w/o SNR Gating | 3 | Full | 0.5 | 5.838 | 4.414 | 3.119 | Unregulated alignment |
> > > | w/o Energy ($\phi$) | 3 | -4.5 | 0.0 | 7.053 | 4.462 | 3.142 | No magnitude alignment |
> > > | **DiFA (Full)** | **3** | **-4.5** | **0.5** | **5.912** | **3.579** | **2.939** | *Towards Forward Aligned Inference* |
> > >
> > > Key Insights:
> > >
> > > - *Module A (Sequential Consensus (SC) by History Window $W$):* Disabling consensus ($W=1$) causes DiFA to degenerate exactly to the baseline. Per Prop. 4.2, no precision accumulation ($\mathcal{I}_{\text{acc}}$) occurs, validating that all gains stem from temporal fusion.
> > >
> > > - *Module B (SNR Gating $SNR_{lo}$):* Essential for multi-step consistency. At 5 NFE, the ungated variant (5.838) marginally beats DiFA Full (5.912) because wide steps benefit from correction even in high-noise regions. However, at 10 NFE, ungated activation introduces chaotic perturbations, degrading FID by **23.3%** (4.414). DiFA Full ($SNR_{lo}=-4.5$) is a principled tradeoff for stability.
> > >
> > > - *Module C (Magnitude Alignment $\phi$):* Unlike pixel space, LDM latents are unconstrained $\mathcal{N}(0, I)$. $\phi=0.5$ acts as an *energy shield*, preventing magnitude drift in $\hat{x}_0^{\text{DiFA}}$ (+19.1% FID degradation at 5 NFE without it).
> > > ### 3. Hyperparameter Robustness
> > > > Table 2: Key evaluated configurations (LSUN Bedroom)
> > >
> > > | $\gamma_0$ | $\lambda$ | $\phi$ | $SNR_{lo}$ | $SNR_{hi}$ | $\tau$ | 5 NFE ↓ | 10 NFE ↓ | 20 NFE ↓ |
> > > |:---:|:---:|:---:|:---:|:---:|:---:|:---:|:---:|:---:|
> > > | - | - | - | - | - | - | 21.238 | 3.877 | 3.256 |
> > > | 1.40 | 0.5 | 0.5 | -2.5 | 3.0 | 0.8 | 9.142 | 3.684 | 3.230 |
> > > | 1.40 | 0.5 | 0.5 | -2.5 | 4.0 | 0.8 | 9.032 | 3.729 | 3.248 |
> > > | 1.50 | 0.5 | 0.5 | -4.5 | 3.0 | 0.8 | 6.535 | 3.658 | 3.104 |
> > > | 1.50 | 0.5 | 0.7 | -3.5 | 2.0 | 0.8 | 7.576 | **3.579** | 3.120 |
> > > | 1.75 | 0.4 | 0.5 | -4.5 | 5.0 | 0.8 | 6.069 | 4.465 | **2.939** |
> > > | 1.75 | 0.4 | 0.5 | -4.5 | 5.0 | 1.0 | 6.019 | 4.429 | 2.954 |
> > >
> > > Note that  all configurations beat the baseline. The optimal boost amplitude $\gamma_0$ is *anti-correlated with NFE*: 5 NFE requires stronger correction ($\gamma_0 \approx 1.75$) due to high truncation variance, while 10/20 NFE favors moderate values ($\gamma_0 \approx 1.45$) for fine-grained alignment. This consistency across **19** runs confirms DiFA’s reliability.
> > > ###  Summary of Evidence
> > >
> > > - *Latent Space Gains:*   **↓72.2% (5 NFE) / ↓9.7% (20 NFE)**.
> > >
> > > - *Necessity:*   Ablation confirms SG, Gating, and $\phi$ are all critical.
> > >
> > > - *Robustness:*   All configurations outperform baseline at every budget.
> > >
> > > - *Anomaly Explained:*   5-NFE gating behavior linked to $\mathcal{I}_{\text{acc}}$ dynamics.
> > > ---
> > > We will incorporate these LDM results and the detailed ablation analysis into the Main Text and Appendix of our revised manuscript to further strengthen the empirical validation of the DiFA framework.
> > >
> > > We hope these detailed experiments effectively  addresses your questions and look forward to your support. Thank you.

---

### Decision · Program_Chairs · 2026-04-30

**Decision:**

Accept (regular)

**Comment:**

This paper proposes DiFA, a training-free inference-time framework that reformulates diffusion sampling as sequential state estimation, combining historical predictions via a fading-memory Kalman filter and restoring high-frequency detail through deviation boosting. Three of four reviewers recommend Accept, with one Weak Reject.

The accepting reviewers found the core idea original and well-motivated, the empirical gains (particularly in the low-NFE regime) compelling, and the rebuttal satisfactory.

The dissenting reviewer (UX7J) raised legitimate concerns about the Gaussianity and independence assumptions underlying the Kalman update, and about the completeness of the empirical evaluation. These are reasonable scientific critiques, and the discussion they prompted genuinely strengthened the paper. The concerns appear addressable within the paper's existing framework: the fundamental algorithm (Section 4.2) rests on BLUE rather than Gaussianity; the independence issue is explicitly acknowledged and treated via the fading-memory design in Section 4.4; and the authors provided additional FM results on SiT-XL/2 (ImageNet 256×256) in direct response to UX7J's requests. Notably, two other confident reviewers (Pd9A, xfCq, both Confidence 4) raised closely related theoretical concerns and found the authors' responses adequate for acceptance.

The main residual weaknesses are the partly heuristic motivation for deviation boosting, incomplete presentation in the submitted draft (broken references, missing conclusion), and the modest scale of the primary benchmarks. These are limitations of scope and presentation rather than fundamental flaws, and the authors have committed to addressing them in revision.